# Substrate accessibility regulation of human TopIIa decatenation by cohesin

Erin E. Cutts [1,4], Sanjana Saravanan [1], Paul Girvan [2,3], Benjamin Ambrose[2,3], Gemma L. M. Fisher [1], David S. Rueda [2,3] ✉ & Luis Aragon [1] ✉

Human topoisomerase II alpha (TOP2α) resolves DNA intertwines between sister chromatids during mitosis. How cohesin, an SMC complex that holds sister chromatids, affects TOP2α decatenation is unclear. To addres this, we developed a quadruple-trap optical tweezers assay to create DNA braids and study TOP2α decatenation at the single-molecule level in real-time. We show that TOP2α resolves both single and multiple braids but becomes inefficient at forces exceeding 28 pN. TOP2α is sensitive to DNA geometry, exhibiting a chiral preference for right-handed braid crossings, and requires loading directly at DNA crossovers to act. Pre-loading TOP2α onto individual DNA strands before braid formation, in the presence of ATP, prevents decatenation. Finally, we show that human cohesin, but not condensin I, binds stably to DNA braids and blocks TOP2α activity. Our study provides novel insights into the role of substrate accessibility in regulating TOP2α's activity and highlights cohesin as a barrier to decatenation.

The double helical structure of DNA[1] provides stability to the molecule, however when topologically constrained, it can also generate entanglement due to the winding of the DNA molecule. For example, replication fork progression and parental DNA strands separation result in positive and negative supercoiling, and intertwines or catenanes between replicated daughter DNA molecules[2,3]. Topological entanglements such as these need to be removed before chromosomes can segregate during cell division[4]. Failure to remove these results in cell death. Hence, cells have evolved a class of enzymes, DNA topoisomerases, which alleviate intertwines and torsional strain by introducing transient breaks on the DNA[5].

There are two main types of topoisomerases, termed type I and type II, depending on the number of DNA strands they cleave and whether they require ATP[6]. Type I topoisomerases cleave only one DNA strand, while type II topoisomerases are characterized by their ability to cleave both DNA strands and use the energy derived from ATP hydrolysis. While both type I and type II can alleviate the build-up of torsional strain due to supercoiling, only type II can resolve intertwining between two DNA molecules[6].

Type II topoisomerases are functional dimers, with a structure composed of three gate regions (Fig. 1a). The N-gate binds and hydrolyses ATP, the DNA-gate cleaves DNA, and the C-gate regulates DNA release. Two segments of DNA are bound: the gate segment (G-segment) and the transfer segment (T-segment). A transient double-stranded DNA break is made in the G-segment by a trans-esterification reaction covalently linking the 5′-ends of each strand with a pair of conserved tyrosine residues allowing the T-segment to pass through the G-segment[7] (Fig. 1b).

Human cells contain two isoforms of type II topoisomerase (TOP2), TOP2α and TOP2β[8], primarily differing in the polypeptide sequence of their carboxy-terminal domains (CTDs). TOP2α has been associated with DNA replication and mitosis[9], whereas TOP2β has been linked to transcriptional regulation of gene expression in differentiated cells[10,11]. TOP2α is cell-cycle regulated, increasing in activity from mid-S phase until mitosis and decreasing rapidly upon mitotic completion[12], and thus, thought to be the main enzyme removing DNA intertwines during mitosis. It acts along the chromosome arms prior to metaphase[13,14] and at centromeres during the onset of anaphase[15].

[1]DNA Motors Group, MRC Laboratory of Medical Sciences, Du Cane Road, London W12 0HS, UK. [2]Single Molecule Imaging Group, MRC Laboratory of Medical Sciences, Du Cane Road, London W12 0HS, UK. [3]Department of Infectious Disease, Faculty of Medicine, Imperial College London, Du Cane Road, London W12 0HS, UK. [4]Present address: School of Biosciences, Faculty of Science, University of Sheffield, Sheffield S10 2TN, UK. ✉e-mail: david.rueda@imperial.ac.uk; luis.aragon@lms.mrc.ac.uk

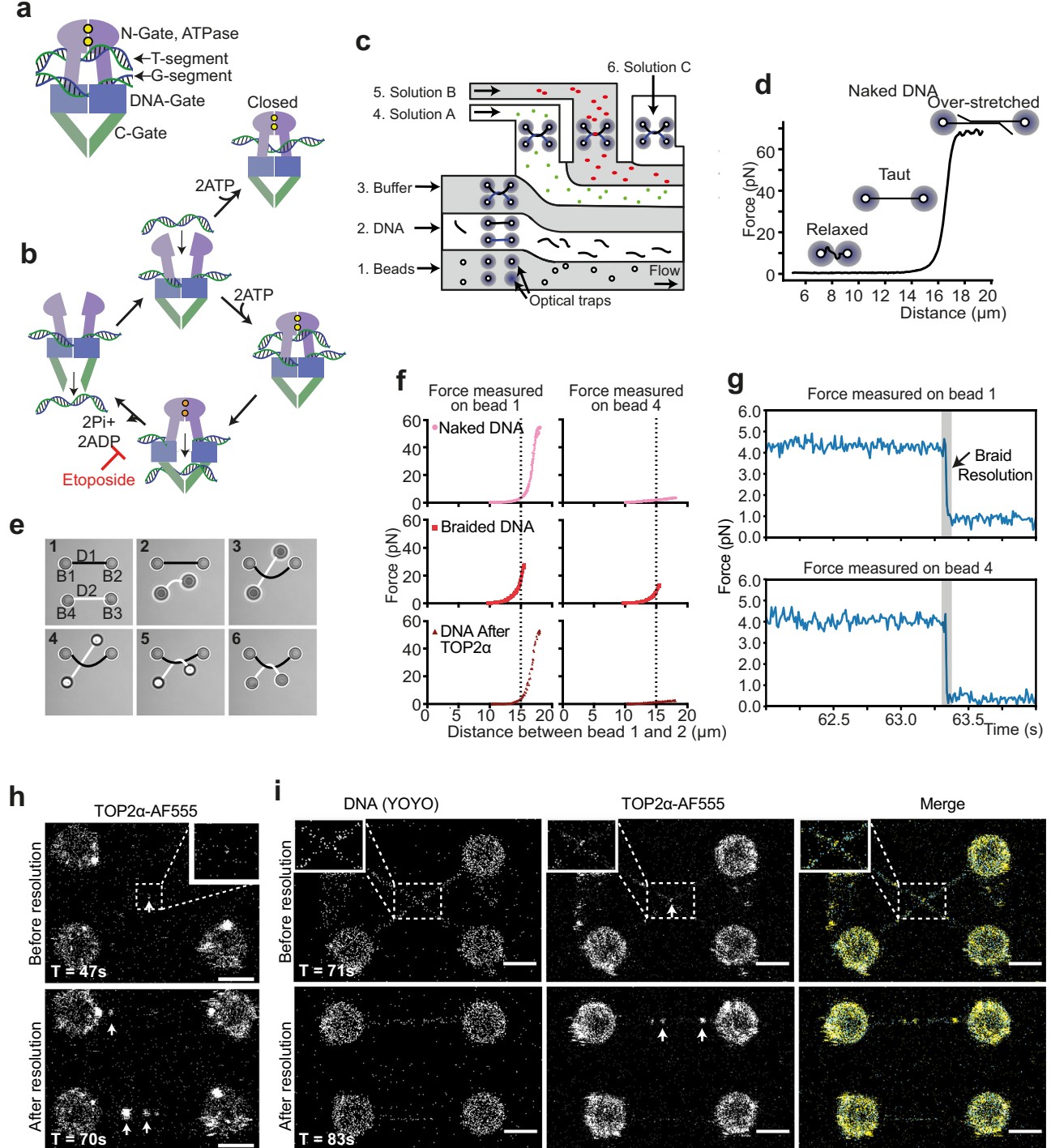

**Fig. 1 | Single-molecule visualisation of TOP2α DNA braid resolution.**
**a** Schematic of TOP2α enzyme with two DNA segments. **b** Diagram showing current understanding of TOP2α enzymatic mechanism. **c** Schematic of microfluidic flow cell used for single-molecule assays. Four beads can be captured by optical traps in channel 1, a DNA can be caught between pairs of beads in channel 2, force extension curve collection and DNA braid is performed in channel 3 and different experimental solutions can be added to channels 4, 5 and 6. **d** Example of force extension curve of naked DNA following Worm-Like Chain model, indicating DNA conformation at different forces. **e** Images from bead movie (Supplementary Movie 1) with drawn DNA illustrating how a DNA braid is produced. DNA1 and 2 are held between beads 1 and 2, and 3 and 4, respectively. Beads 3 and 4 are moved down in Z, bead 3 is passed under DNA1, then moved up in Z before being passed over DNA1. **f** Force extension curves in the absence of YOYO, pulling on DNA 1 to increase distance between bead 1 and 2, measuring force experienced by bead 1 or 4. Pulling on naked DNA 1 results in force on bead 1 and minimal force on bead 4. After

braiding, DNA1 appears to have a shorter contour length with respect to bead 1 to 2 distance (as indicated by dotted line) and increase in distance between bead 1 and 2 results in an increase in force of bead 4. After TOP2α treatment, DNA returns to that of naked DNA. **g** Force vs time curve of the braid resolution event shown in (**h**).
**h** Direct visualisation of braid resolution by TOP2α-AF555, after 47 seconds, before resolution, TOP2α-AF555 is in the centre of the 4 beads, and after 70 seconds, after resolution, TOP2α-AF555 is found between beads 1–2 and 3–4. Representative images shown. Similar results were observed in 23 out of 26 independent experiments. Representative image showing observations in **i**. **i** Direct visualisation of braid resolution by TOP2α-AF555 in the presence of YOYO, after 71 seconds, before resolution, braid can be observed as a cross using YOYO DNA dye, with TOP2α-AF555 at the junction and after 83 seconds, after resolution parallel DNA with bound TOP2α-AF555 can be observed. White arrows indicate bound TOP2α-AF555, in all cases, the scale bar is 4 μm. Representative images shown, all 8 independent experiments showed similar outcomes (N = 8/8).

TOP2β is enriched at boundaries of topologically associated domains (TADs)[16], suggesting that it resolves topological stress arising during genome folding in interphase nuclei.

Before the discovery that cohesin complexes affect sister chromatid cohesion[17,18], intertwines were thought to be the main means of sister chromatid cohesion, maintaining the physical proximity of chromatids until mitosis[19]. Interestingly, cohesin-bound sites are also thought to contain sister chromatid intertwines (SCIs)[20,21]. However, it is unclear whether the presence of cohesin at these sites promotes catenation or inhibits decatenation of SCIs by TOP2 during interphase. Consistent with this view, TOP2α-dependent decatenation of centromeric regions in HeLa cells occurs only after cohesin is removed by separase at the onset of anaphase[15].

In this study, we reconstitute the activity of human TOP2α at intertwines in vitro. First, we develop a single-molecule assay[22–26] to generate braided DNA molecules that are held under force to mimic intertwines under spindle forces during mitosis. We then show that these can be resolved by TOP2α. We use this approach to demonstrate how pulling force, substrate structure, and the sequence of binding substrate and ATP affect TOP2α activity. We further demonstrate that cohesin efficiently binds intertwined DNA and inhibits TOP2α resolution, suggesting that cohesin may associate with intertwines formed during replication and protect centromeric intertwines from TOP2α until its removal in metaphase/anaphase.

## Results

### Generating DNA braids to study TOP2α activity in the optical tweezers

To study the activity of human TOP2α resolving intertwines, we used a correlative quadruple-trap optical tweezers system with confocal fluorescence detection (Q-Trap)[22–26]. Experiments were performed in a multi-channel laminar flow cell, with up to six defined channels. Channels 1 and 2 contained streptavidin-coated polystyrene beads and λ-DNA molecules (48.5 kb) with biotinylated ends on one strand, respectively, whereas channels 3 to 6 contained different buffers and proteins depending on the experiment (Fig. 1c).

The capture of a single λ-DNA between bead pairs was confirmed by collecting force-extension (FE) curves by mechanically stretching the DNA while measuring the resultant force (Fig. 1d). The FE curve of DNA has a characteristic shape, exhibiting a sharp increase in force as the DNA contour length is reached (-16 μm for 48.5 kb long λ-DNA, Fig. 1d), which can be described by the Worm-Like Chain model.

To generate braided DNA substrates, we used a previously established approach[22–24,26]. Two pairs of beads (B1/B2 and B3/B4) are initially trapped in channel 1 using the four optical traps and then transferred to channel 2, where two DNA molecules (D1 and D2) are captured between the respective bead pairs (Fig. 1e and Movie 1). The two DNA molecules are then moved to channel 3 containing buffer, where FE curves are measured to confirm the presence of one intact λ-DNA between each bead pair. A scripted series of movements is used to generate a braid between D1 and D2: first the two DNA molecules are placed parallel to each other on the same Z-plane (Fig. 1e, frame 1). D2 is then moved down in the Z-plane and B3 is passed under D1 (Fig. 1e, frames 2-3). D2 is then raised above D1 in the Z-plane (Fig. 1e, frame 4), B3 is moved over D1 (Fig. 1e, frame 5), and finally, B3 and B4 are lowered back into the initial Z-plane, such that both D1 and D2 are in the focal imaging plane (Fig. 1e, frame 6) and beads are aligned to form a rectangle. This sequence of manipulations results in a single braid between the two captured DNA molecules that contains a right-handed crossing. Since D1 and D2 are torsionally relaxed and free to rotate because they are attached to the streptavidin beads through biotin on one of the DNA strands, this protocol produces braids with similar characteristics to CatA-type catenanes in plasmids, albeit with larger crossing angles[27].

The formation of braids can be confirmed from changes in FE curves, as before braiding, FE curves between B1 and B2 show the expected Worm-Like Chain profile, characteristic of λ-DNA molecules (Fig. 1f, Supplementary Fig. 2b; Naked DNA), while after braiding, the DNA contour length appears shorter (Fig. 1f, Supplementary Fig. 2b; Braided DNA) and force can be detected on B4, i.e., an increase in force can be observed on D2 while pulling on D1, as expected from the coupling of the two DNA molecules.

### Visualising TOP2α braid resolution

To directly visualise TOP2α braid resolution, human TOP2α was purified and labelled with Alexa Fluor 555 (TOP2α-AF555) (Supplementary Fig. 1a). Bulk DNA decatenation assays were used to demonstrate that labelled material was active (Supplementary Fig. 1c). Labelling efficiency ranged from 30-70% across different purifications, resulting in a minimum of 50% of dimers having at least one fluorophore.

DNA braid resolution was observed when a braid (held at 5 pN) was incubated in channel 4 containing 2 nM TOP2α-AF555 and 1 mM ATP. Resolution could be observed as a sharp, simultaneous drop in the force measured on B1 and B4, exerted by D1 and D2, respectively (Fig. 1g, and Supplementary Fig. 2c) and via direct imaging (Fig. 1h, Supplementary Fig. 2a). TOP2α-AF555 bound double-stranded DNA and often appeared briefly at the middle of the 4 beads, where the braid junction was expected (visible at the junction 14 times out of 26 cases, Fig. 1h; scan time 47 s, Supplementary Fig. 2a), and could be observed between B1-2 or B3-4 after resolution, once DNA tension was applied to move it back into the imaging plane, suggesting that TOP2α-AF555 was bound to the resolved DNA (Fig. 1h; scan time 70 s, Supplementary Fig. 2a, and Supplementary Movie 2). In the cases where TOP2α-AF555 was not visible, we assume the event either occurred too quickly to observe or was performed by an unlabelled/photobleached TOP2α. The FE curve shape returned to that characteristic of naked DNA after incubation with TOP2α-AF555, further confirming braid resolution (Fig. 1f; DNA after TOP2α, Supplementary Fig. 2b).

We also followed TOP2α-AF555-mediated resolution of the two DNAs by imaging DNAs in the presence of YOYO-1 dye (Fig. 1i, and Supplementary Fig. 2d)[28]. We could visualize the braid between D1 and D2, which appeared as a cross in the middle of the four beads (Fig. 1i, scan time 71 s and Supplementary Fig. 2d; before decatenation). This structure was rapidly resolved into two fully separated DNA molecules (Fig. 1i; scan time 83 s, Supplementary Fig. 2d, Supplementary Movie 2) consistent with resolution by TOP2α. Importantly, in many cases, we were able to visualize the binding of TOP2α-AF555 to the DNA junction just ahead of its resolution (Fig. 1i, Supplementary Fig. 2d). We thus conclude that TOP2α efficiently resolves a single DNA braid generated between the pair of λ-DNA molecules in our experimental setup.

### Effects of Force on TOP2α resolution

Our aim was to examine TOP2α activity in the context of mitosis, where spindle forces are present. Previous studies have examined the effects of force on TOP2α rate[29,30] finding that the rate reduced with increased tension. However, these studies were performed at forces less than or equal to 2.5 pN, and as maximal mitotic forces at centromeric regions could exceed 2.5 pN[31], we tested TOP2α activity over a higher force range.

After generating a single DNA braid in the buffer channel, beads were arranged in a rectangle, with a fixed distance between bead 1 and 4 of 7.8 ± 0.3 μm (mean ± SD), varying the distance between beads 1 and 2, and beads 3 and 4, to achieve fixed forces of 5, 15, 25, 30 and 45 pN. We analysed the resolution of braided DNAs by TOP2α-AF555 as a Bernoulli trial, using a 5-minute sampling window, to determine the mean resolution probability at each force (Fig. 2a, b). Contrary to studies performed at less than 5 pN where TOP2α-AF555 activity was strictly force dependent[29,30], we found that the resolution probability did not significantly change between 5 and 15 pN (mean resolution

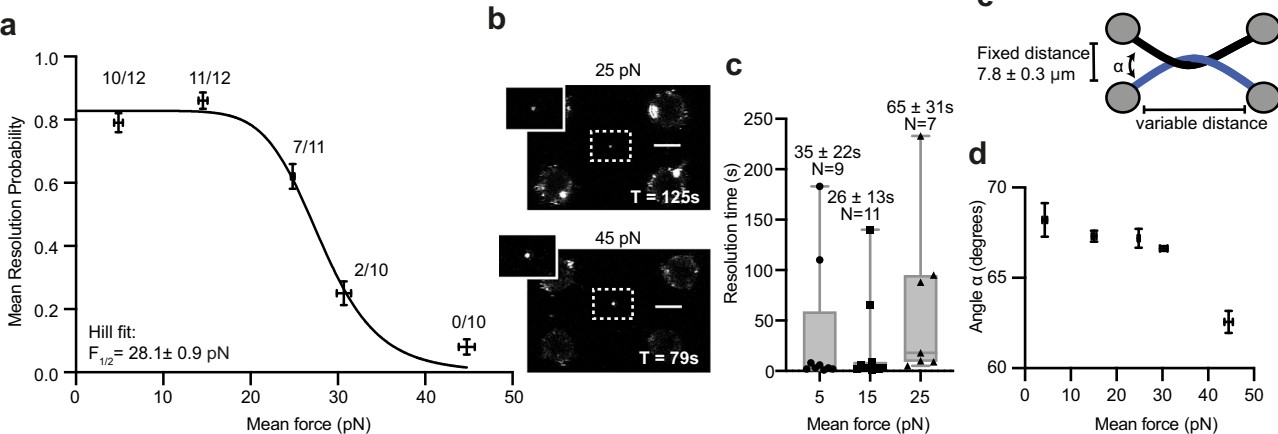

**Fig. 2 | The effect of force on TOP2α resolution. a** Mean resolution probability of TOP2α-AF555 acting on DNA held at different forces, modelled as a Bernoulli process, with 5-minute sampling time. Number of resolution events out of total number of technical replicates is indicated above each force condition (5, 15, 25, 30 and 45 pN); error bars indicate one standard error. Source data are provided as a Source Data file. **b** Example images from failed resolution event at 25 and 45 pN. **c** Resolution time after entering the protein channel at different mean forces. N refers to the number of molecules tested in different technical replicates. Box plot whiskers represent the full data range (minimum to maximum), boxes indicate the interquartile range (25th to 75th percentiles), and the central line marks the median. Source data are provided as a Source Data file. **d** Estimated crossing angle, α, at different mean forces (5, 15, 25, 30 and 45 pN). The total number of technical replicates per force was as in (**a**) (5 pN: n = 12, 15 pN: $n$ = 12, 25 pN: $n$ = 11, 30 pN: $n$ = 10 and 45 pN: $n$ = 10, pN). Error bars indicate standard error. Source data are provided as a Source Data file. **e** Schematic depicting crossing angle, α, between the two braided DNAs.

probability ± standard error of 0.79 ± 0.03 and 0.86 ± 0.03, respectively, or N = 10/12 and N = 11/12, respectively). However, the resolution probability decreased at 25, 30 and 45 pN, with resultant mean resolution probabilities of 0.62 ± 0.13, 0.25 ± 0.12 and 0.08 ± 0.08, respectively, (or N = 7/11, N = 2/10 and N = 0/10, respectively). Hence, the resolution exhibited a half stall force of $F_{1/2}$ = 28.1 ± 0.9 pN (fit ± SE).

As previous studies examined rates rather than resolution probability, we then determined the time between entering the channel and the DNA braid resolving at forces where resolution occurs. For 5, 15 and 25 pN, the mean resolution times ± SE were 35 ± 21, 26 ± 13 and 65 ± 31 s, respectively (Fig. 2c). Our time measurement includes both time of association and time of resolution, however acquiring a direct enzymatic rate would require a faster imaging rate and a large data set, which is not feasible with our system. Collectively, these results suggest that the sharp drop in resolution probability observed in our assay at higher tensions is better explained by the presence of DNA distortions—likely preventing G-segment cleavage or T-segment capture—rather than by a progressive decrease in resolution rate with increasing tension.

To ensure that the effect in resolution probability observed is not due to changes in crossing angle at the different tensions, we estimated the crossing angle, α, based on the bead positions (Fig. 2d, e, Supplementary Fig. 3a-c), assuming equal DNA length, and found no significant differences at 5, 15, 25 and 30 pN, with the crossing angles of 68.2 ± 0.9, 67.3 ± 0.3, 67.2 ± 0.5, 66.6 ± 0.1 degrees, respectively (mean ± SD, Fig. 2d). There was a slight reduction in crossing angle at 45 pN to 62.5 ± 0.6 degrees, but given the extent of the change in crossing angle and as it occurred at forces higher than F1/2 = 28.1 ± 0.9 pN, it is unlikely to have caused the observed drop in resolution probability. This angular range is compatible with the TOP2α favoured bend angle of ~120 degrees[32,33].

### TOP2α resolution of DNA braids requires ATP hydrolysis
Having established that our approach yields a robust measure of TOP2α activity at forces of 15 pN and lower, we tested our assay against factors known to affect TOP2α activity.

Previous studies have demonstrated that ATP hydrolysis is required for efficient decatenation by TOP2α. ATP analogues have been shown to allow T-segment transfer to the C-gate[34], however

T-segment release from the C-gate[35] requires prior re-ligation of the G-segment.

We used 1 mM ATPγS, a non-hydrolyzable ATP analogue, in our assays instead of ATP and observed no resolution events (N = 0/14, Fig. 3a), despite observing that TOP2α-AF555 localised at the DNA braid junction in all cases (N = 14/14, Fig. 3b, and Supplementary Fig. 4a). Since we observed binding to the braid junction, we assume that treatment with the ATP analogue does not close the N-gate prior to DNA association, which has been shown to block association with non-linear DNA[36]. In the presence of ATPγS, TOP2α-AF555 was visible for multiple frames, allowing 2D Gaussian fitting of the fluorescence intensity. Fitting the histograms of fluorescence intensity for the dataset suggests two populations, one at a mean photon count of 3.5 ± 1.0 (mean ± SD) and a second with a mean photon count of 6.8 ± 0.4 (Fig. 3c). Hence, the fluorescence intensities and populations are consistent with that expected for singly and doubly labelled TOP2α-AF555 dimers, confirming single-molecule conditions.

Consistent with our ATPγS data, the use of 1 mM of the ATP analogue AMPPNP also resulted in no resolution (N = 0/8, Fig. 3a, Supplementary Fig. 4b). Based on previous studies[34,35], the resolution failure could be caused by the T-segment DNA being trapped in the C-gate. To determine whether DNA trapped within the C-gate prevents resolution in our assay, we used the chemotherapy agent etoposide, which binds to the cleaved G-segment, preventing re-ligation, while allowing a single round of T-segment transfer. Since the G-segment is re-ligated before the T-segment is released from the C-gate[35], etoposide should also trap the T-segment DNA in the C-gate under native conditions. We found that addition of 50 μM etoposide inhibited resolution, resulting in a mean resolution probability of 0.47 ± 0.03 (N = 6/13), while still localising to the braid junction in all cases where resolution was inhibited (N = 7/7 of unresolved repeats, Supplementary Fig 4c). Collectively this data suggests that our assay requires release from the C-gate in order to visualise resolution.

### TOP2α binds to DNA Crossovers for Efficient Decatenation
Large amounts of TOP2α associate with chromosomes in mitotic cells, and TOP2α is proposed to have non-catalytic roles in chromosome structuring, as Auxin-induced degradation of TOP2α does not phenocopy its inhibition[37,38]. We frequently observed TOP2α binding to

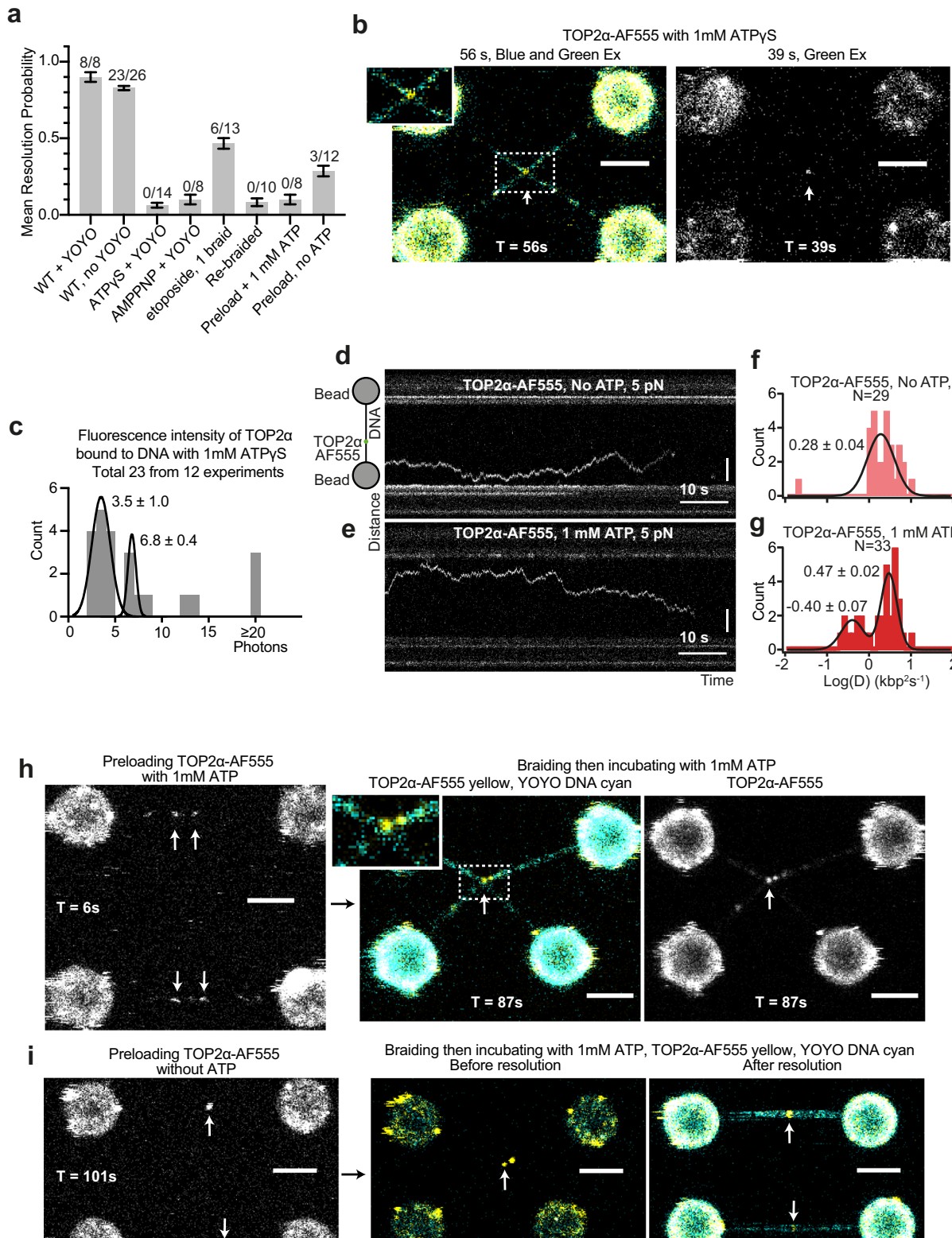

double-stranded DNA (Fig. 1h and i, arrows) in addition to the braided junction. Therefore, we aimed to determine whether TOP2α pre-bound to dsDNA could resolve intertwines.

First, we confirmed that TOP2α can load onto dsDNA molecules (Fig. 3d) by incubating a single piece of dsDNA in a channel with TOP2α-AF555. Kymograph analysis showed that TOP2α-AF555 molecules not only bind but also freely diffuse along dsDNA. This was

observed both in the presence and absence of ATP (Fig. 3d and e) with fluorescence intensity and photobleaching steps consistent with TOP2α dimers (Supplementary Fig. 4d). TOP2α in the absence of ATP had a single diffusion coefficient of $1.9 \pm 0.2$ kbp$^2$s$^{-1}$ (Fig. 3f and Supplementary Fig. 3e and f, N = 29), while in the presence of 1 mM ATP there were two populations, with diffusion coefficients of $0.4 \pm 0.1$ kbp$^2$s$^{-1}$ and $3.0 \pm 0.2$ kbp$^2$s$^{-1}$ (Fig. 3g, N = 33). The presence of a more

**Fig. 3 | Interrogating the TOP2α enzymatic cycle. a** Mean resolution probability of DNA braids by TOP2α-AF555 in different conditions. Number of resolution events out of total number of technical replicates is indicated above each condition, error bars indicate one standard error. Source data are provided as a Source Data file. **b** Example image of TOP2α-AF555 in the presence of 1 mM ATPγS, with blue and green excitation and only green excitation. **c** Histogram of fluorescence intensity of TOP2α-AF555 bound on braided DNA. Fit shown with mean and sd of Gaussian. Source data are provided as a Source Data file. **d, e** Kymographs showing TOP2α-AF555 associating and diffusing on DNA held at 5 pN in the absence and presence of ATP, respectively. **f, g** Quantification of the diffusion coefficients from kymographs with and without 1 mM ATP, respectively. Mean and sd of Gaussian fit indicated. Source data are provided as a Source Data file. **h** Representative images of TOP2α-AF555 pre-loaded on dsDNA in the presence of 1 mM ATP (left), prior to generating a braid and moving into a channel with 1 mM ATP and 0.1 nM YOYO. Image shown with both TOP2α-AF555 and YOYO channels (middle) or just TOP2α-AF555 channel (right). All 8 independent replicates yielded similar results (N = 8/8). **i** Representative images of TOP2α-AF555 pre-loaded on dsDNA in the absence of ATP (left), prior to generating a braid and moving into a channel with 1 mM ATP and 0.1 nM YOYO. Image shown with both TOP2α-AF555 and YOYO channels, before (middle) and after (right) resolution. Scale bar is 4 μm in all cases. The resolution outcome shown was observed in 3 independent experiments (total N = 12).

slowly diffusing population suggests ATP induces a conformational change slowing down diffusion, however, this diffusion is still greater than non-diffusing enzymes, such as endonuclease I bound at a holiday junction, which has a diffusion coefficient of $0.01 \pm 0.01$ kbp$^2$s$^{-1}$ [39].

We then introduced a DNA braid after loading TOP2α, either in the presence or absence of ATP, followed by incubation with YOYO and 1 mM ATP. We found that loading TOP2α on DNA in the absence of ATP permitted some resolution events, with a mean resolution probability of $0.29 \pm 0.03$ (N = 3/12, Fig. 3a and h). However, no resolution was observed when TOP2α was loaded in the presence of ATP (N = 0/8, Fig. 3a), despite TOP2α-AF555 being visible at the junction in most cases (N = 6/8, Fig. 3i). Similarly, no resolution occurred if DNA was re-braided after a round of resolution (N = 0/10, Fig. 3a and Supplementary Fig 4g).

Across different conditions, we estimated the DNA crossing angle, and found no significant difference (Supplementary Fig. 3d). Collectively, these data suggests TOP2α-AF555 can bind DNA and the ATPase gate can close in the absence of the T-segment (Fig. 1a and b), consistent with previous data[34]. This also implies that TOP2α must bind at or very near the braid junction to efficiently resolve DNA, aligning with previous findings that suggest TOP2α preferentially associates with DNA crossover[40].

## TOP2α resolves DNAs with multiple braids

Previous work studying TOP2α activity using single-molecule approaches has demonstrated that individual TOP2α dimers can process multiple DNA crosses in supercoiled and catenated templates in a processive manner when subjected to low forces[30,41,42]. We next tested whether higher numbers of braids could also be resolved efficiently in our assay by repeating the DNA braiding procedure four times (Fig. 4a). We observed that the efficiency of resolution of DNAs with four braids was comparable to that of DNAs containing a single braid (mean resolution $0.83 \pm 0.01$ for one braid, vs $0.82 \pm 0.02$ for four braids) (Fig. 4b). To confirm that TOP2α was proceeding through the enzymatic cycle multiple times in our assay, we added etoposide, which only inhibits the step of the catalytic cycle when the G-segment is cleaved. In the presence of 50 μM etoposide, the mean probability of resolution dropped from $0.47 \pm 0.01$ for one braid (6 resolution events out of N = 13, Fig. 3a) to $0.09 \pm 0.03$ for four (0 out of 9 cases, Fig. 4b and c), consistent with the expected probability of a single resolution being inhibited four times.

Interestingly, we were able to detect either one or two drops in force, when monitoring the traps during the resolution of multiple braids (Fig. 4d-e). When resolution exhibited one drop in force, in most cases, there was a consistent fluorescence intensity at the junction (Fig. 2d, 9 of 13 resolution events). However, in three cases, we detected two drops in force which was coupled with an increase in TOP2α fluorescent intensities over time suggesting the action of more than one TOP2α dimer (Fig. 4e).

In order to estimate how many TOP2α dimers were present in each case, we analysed the fluorescence intensity of TOP2α bound by fitting 2D gaussians. The distribution of photons resulted in two peaks, with mean $\pm$ SD of $2.6 \pm 0.6$ and $5.9 \pm 1.1$ photons (Fig. 4f, Supplementary Fig. 4h,i). The fluorescence intensity quantification suggests that resolutions detected with one drop in force occur by a single-bound complex acting processively in a rapid "burst" that resolves the four braids (Fig. 4d), as it has been demonstrated in previous studies[41,42]. The resolution events involving two drops in force corresponded to cases where more than one TOP2a dimer bound in separate steps, providing evidence that multiple TOP2α can also work together (Fig. 4e). The time resolution of our instrument did not allow us to detect the removal of individual braids in the resolution bursts.

TOP2α is known to be able to processively resolve both right-handed and left-handed DNA braids, with the resolution being faster for right-handed crosses at forces < 0.4 pN[29]. When generating braids in our system, the movement of bead B3 dictates the handedness. When B3 is first moved under D1, the resulting cross is right-handed, while when B3 is first moved over D1 the resulting cross is left-handed (Fig. 4a). TOP2α was able to remove a single left-handed braid, with a mean resolution probability of $0.58 \pm 0.03$ (N = 10/17, Fig. 4b and h), indicating lower efficiency than for right-handed crosses. This is despite the resolution time after entering the TOP2α channel being comparable for right- and left-handed braids ($49 \pm 8$ and $48 \pm 13$ s, respectively, Fig. 4g) and no significant difference in DNA crossing angle (Supplementary Fig. 3e, respectively). Increasing the number of left-handed braids to four resulted in a mean resolution probability of $0.56 \pm 0.03$ (N = 8/14, Fig. 4b), comparable to one left-handed braid (Fig. 4b). Fluorescence intensity quantification revealed two peaks, consistent with one dimer bound at the junction, indicating the processive resolution of left-handed braids (Fig, 4i). No multiple-step resolution events were observed for left-handed braids, despite fluorescence quantification suggesting that more than one dimer was present in some cases.

## Cohesin inhibits TOP2α DNA braid resolution

During mitosis, chromosome individualisation by TOP2 is countered by cohesin-mediated chromosome cohesion[43]. Cohesin forms a tripartite ring structure with compartments that can trap one or two DNA molecules[44–46]. Binding of yeast cohesin to chromosomes correlates with the presence of intertwines between sister chromatids[20,21] and the resolution of catenation by TOP2α has been shown to follow cohesin removal from centromeric regions on mammalian chromosomes[15,47], raising the possibility that cohesin prevents TOP2α from accessing intertwines. With this in mind, we sought to investigate whether the presence of cohesin in our assays is sufficient to affect the resolution of DNA braids by TOP2α.

First, we purified human cohesin (SMC1, SMC3, RAD21 and STAG1-ybbr) and MBP-ΔN-NIPBL and confirmed activity with ATP hydrolysis assays (Supplementary Fig. 1a-b, d). We used the ybbr tag to label the complex with ATTO647N (cohesin-A647N). We then investigated how cohesin interacts with braided DNA. We incubated a right-handed braid with 2 nM cohesin-A647N, 4 nM MBP-ΔN-NIPBL, 1 mM ATP and 0.1 nM YOYO and found cohesin complexes could load onto both

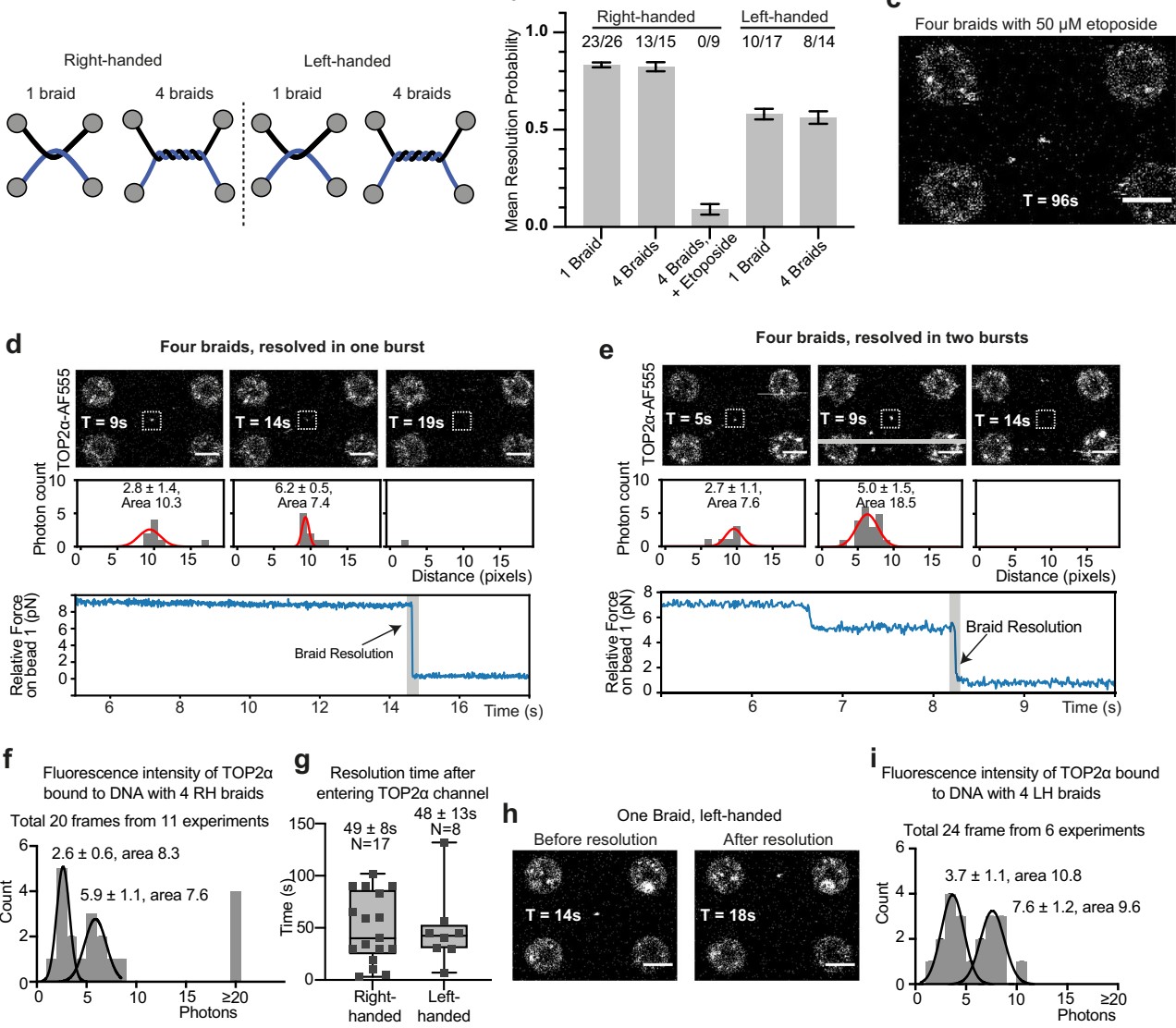

**Fig. 4 | TOP2α DNA braid resolution is processive. a** Schematic of single and four braided, right- and left-handed DNA braids. **b** Mean resolution probability by TOP2α-AF555 of right- and left-handed substrate, and right-handed, four braided substrate with 50 μM etoposide. Data modelled as a Bernoulli process, with 5-minute sampling time. Number of resolution events out of total number of technical replicates is indicated above each condition, error bars indicate one standard error. Source data are provided as a Source Data file. **c** Example image of failed resolution by TOP2α-AF555 in the presence of 50 μM etoposide on a four right-handed, braided substrate. **d**, **e** Force vs time curve of four braid resolution, occurring in one or two bursts, respectively, with corresponding scan images and slice through the TOP2α-AF555 intensity. Grey bar in (**e**) indicates resolution occurring during confocal scan. Further explanation of intensity analysis is provided in Supplementary Fig. 4h. Representative kymographs shown for (**d**), the one-

step transition was observed in 9 independent events (total N = 15), for (**e**), the two-step transition was observed in 3 independent events (total N = 15). **f** Histograms of fluorescence intensities of Gaussian fit of four-braid, right-handed substrates. Source data are provided as a Source Data file. **g** Resolution time of single right- and left- handed braids after entering the protein channel. N refers to the number of molecules tested in different technical replicates. Box plot whiskers represent the full data range (minimum to maximum), boxes indicate the interquartile range (25th to 75th percentiles), and the central line marks the median. Source data are provided as a Source Data file. **h** Example images before and after resolution of left-handed, four braid. **i** Histograms of fluorescence intensities of Gaussian fit of four-braid, left-handed substrates Scale bar is 4 μm in all cases. Source data are provided as a Source Data file.

dsDNA regions as well as the DNA junction (Fig. 5a). Kymographs of cohesin over time suggested that cohesin bound to the DNA braid junction remained immobile, while cohesins bound to dsDNA were able to diffuse, with a diffusion coefficient of $0.7 \pm 0.1$ kbp²s⁻¹ (Fig. 5b, Supplementary Fig. 5a and b). Comparison with the related structural maintenance of chromosome family member, human condensin I, demonstrated that stable binding at the DNA braid is specific to cohesin, as the condensin I complex did not associate stably with the junction as frequently (Supplementary Fig. 5c). Instead, condensin I often dissociated from braids when moved out of the protein channel

(Fig. 5c). After cohesin was bound to the DNA braid, we incubated it with 2 nM TOP2α, 1 mM ATP and 0.1 nM YOYO-1 and tested whether resolution of the DNA braids took place. When pre-incubated with cohesin, resolution by TOP2α was dramatically reduced to a mean resolution probability of $0.25 \pm 0.03$ (3 out of N = 14), compared to $0.90 \pm 0.03$ (8 out N = 8) observed for TOP2α in conditions without cohesin. We also tested the effect of condensin I on TOP2α resolution using the same protocol and observed that mean resolution increased compared to cohesin to $0.57 \pm 0.04$ (7 out N = 12, Fig. 5d, Supplementary Fig. 5d).

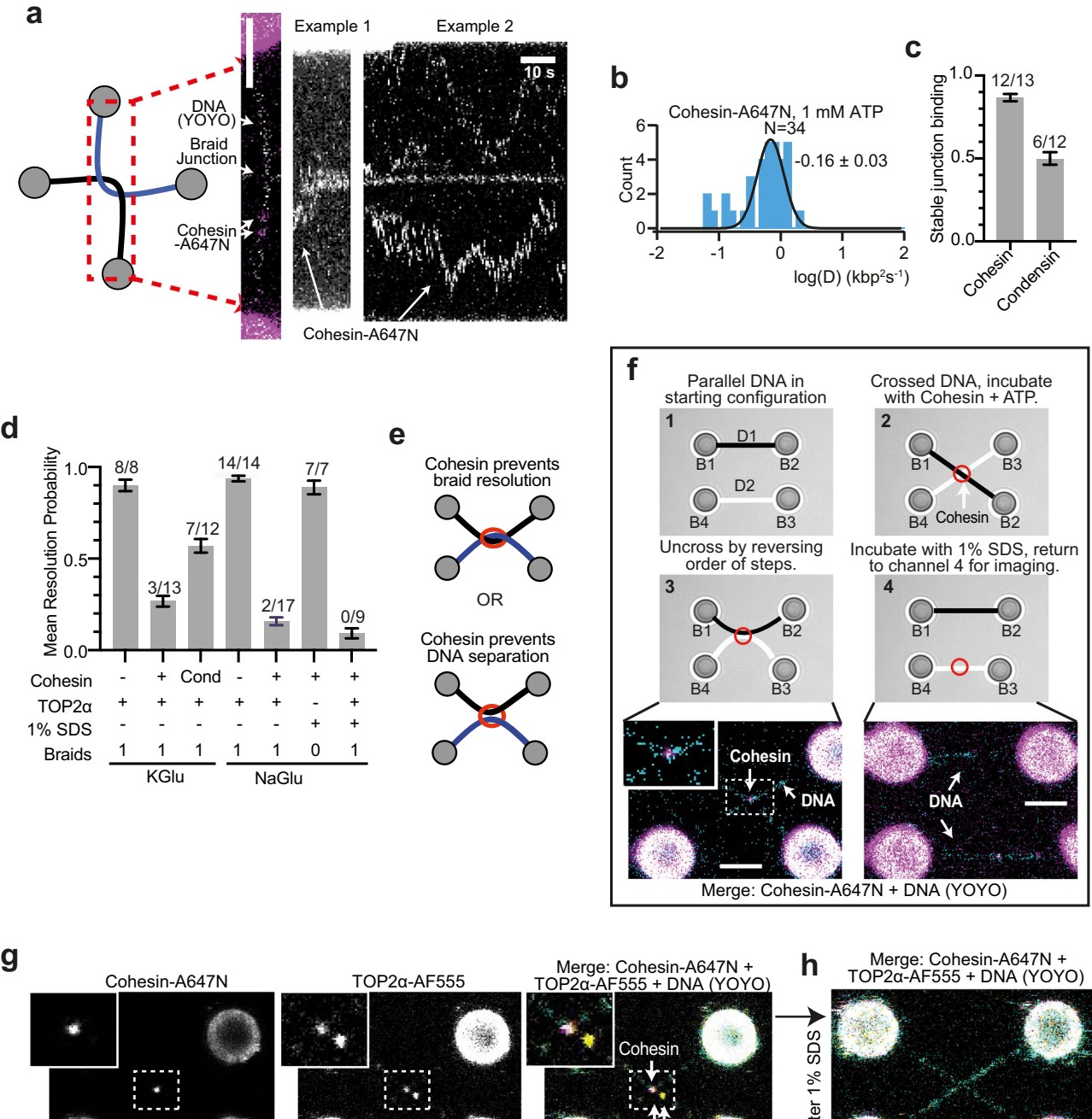

**Fig. 5 | Cohesin blocks TOP2α DNA braid resolution. a** Cohesin-A647N /MBP-ΔN-NIPBL kymograph on braided DNA substrate. Kymographs are generated by reorientating the DNA braid, scanning a limited area and collapsing each scan into one line. **b** Quantification of Cohesin-A647N /MBP-ΔN-NIPBL diffusion coefficients from kymographs on dsDNA. Source data are provided as a Source Data file. **c** Quantification of mean proportion of experiments where cohesin-A647N and condensin I-A647N stably associate with braid junction. Number of resolution events out of total number of technical replicates is indicated above each condition. Data are presented as mean values +/- SEM. Source data are provided as a Source Data file. **d** Mean proportion of DNA resolution in experiments with cohesin-A647N, condensin I-A647N and TOP2α-AF555. Data modelled as a Bernoulli process, with 5-minute sampling time. Number of resolution events out of total

number of technical replicates is indicated above each condition, error bars indicate one standard error. Source data are provided as a Source Data file. **e** Two hypothetical modes of how cohesin could prevent DNA resolution. **f** Schematic of cohesin bridging assay, with representative images of cohesin-A647N DNA bridging from step 3 and 4 shown. DNA1 and 2 (D1 and D2) are crossed by swapping the positions of bead 2 and 3 (B2 and B3) and incubated with cohesin-A647N/MBP-ΔN-NIPBL, 1 mM ATP and 0.1 nM YOYO. B2 and B3 are then returned to original positions, resulting in cohesin bridging D1 and D2. Bridging can be resolved with 1% SDS, before imaging with YOYO. All 7 independent replicates yielded similar outcomes (N = 7/7) (**g**) TOP2α-AF555 localises at a catenation junction with Cohesin-A647N, but is not resolved, even after incubating with 1 % SDS, **h** Scale bar is 4 μm in all cases. Lack of resolution was observed in 15 out of 17 independent experiments.

These results suggest that cohesin specifically binds stably to DNA braids and reduces resolution efficiency by TOP2α. Previous work from our group and others has shown that yeast cohesin can tether two DNA molecules brought into proximity[46,48], raising the possibility that, in our assay, cohesin may prevent separation of the two DNAs by bridging, rather than preventing DNA braid resolution by TOP2α (Fig. 5e). We found that human cohesin-A647N and MBP-ΔN-NIPBL are also able to tether two DNAs and that this bridging is protein mediated, as it can be efficiently dissolved by 1% SDS (resolving 7 out of N = 7) (Fig. 5d and f, Supplementary Movie 3). Therefore, to confirm that cohesin is preventing TOP2α-mediated resolution of the DNA braid, rather than blocking DNA separation by bridging the two DNAs, we tested whether braided DNA incubated with cohesin and TOP2α was sensitive to 1% SDS. We first changed the buffer conditions to use sodium glutamate in place of potassium glutamate, as potassium precipitates with SDS. TOP2α braid resolution and cohesinmediated TOP2α inhibition were similar in buffer containing sodium glutamate, with mean resolution probabilities of $0.94 \pm 0.02$ (14 out of N = 14) and $0.16 \pm 0.02$ (2 out of N = 17), respectively (Fig. 5d). We then incubated single DNA braids sequentially with cohesin, followed by TOP2α, and then 1% SDS, but did not observe DNA resolution, demonstrating that the DNAs remained braided (0 out of 9 cases) (Fig. 5g and h, Supplementary Movie 4). Therefore, we conclude that the presence of cohesin at the DNA braid junction compromises the ability of TOP2α to access and resolve the braid linking D1 and D2.

## Discussion

Numerous studies have investigated the function of TOP2α and TOP2β[49]. Defining the enzymatic details of their activity has contributed significantly to our understanding of their fundamental roles in various biological processes. The majority of studies have examined the rates of decatenation at forces lower than 5 pN, reporting rates of ~3.3 s per cycle at 2.5 pN[29], with resolution rates being sensitive to tension. While the low throughput nature of our experiments and limited time resolution using confocal imaging did not allow us to directly determine a rate, an estimate of 106–200 s per resolution can be obtained assuming a single-exponential process, using the probability of resolution across different wild-type conditions at 5 pN and the 5-minute sampling window. This rate is perhaps slower than expected given a two-fold increase in force, but this estimate represents the association and resolution time, while existing rates are that between sequential cycles. Existing studies differ in the interpretation of the rate limiting step. Charvin et al. suggested, given the force sensitivity, the rate limiting steps of TOP2 must be that where work is done against the pulling force, that is, DNA re-ligation[30], whereas Neuman et al. and Seol et al. suggest T-segment capture is rate limiting, particularly in the case of single DNA braids[29,50]. Our data displayed no significant difference between resolution of one and four braids, nor at 5 and 15 pN of force, instead exhibiting a half-stalling force of $F_{1/2} = 28.1 \pm 0.9$ pN. Given the rate is force sensitive at low forces, this could suggest the rate limiting step changes at higher forces. At lower forces, two DNA segments would be able to fluctuate more hence defined crossing of two DNA segments may be less frequent than at higher forces, while higher forces could result in structural distortions in the DNA, previously observed in optical tweezer entanglements[23], reducing association, T segment capture or cleavage. A similar effect on rate was observed at lower forces for type II topoisomerase Top IV[30]. Previous studies suggest that a single kinetochore microtubule attachment can generate >10 pN and human chromosomes can have up to twenty such attachments[31], therefore, it is possible that forces greater than 30 pN could be generated during mitosis, and TOP2α may fail in such conditions.

DNA conformation and structure could also explain the chiral preference of TOP2α to resolve right-handed DNA braids (this study and ref. 29). Intertwining between sister chromatids is a consequence of the late stages of DNA replication[2,3]. Due to the limited space between converging forks, topoisomerase activity ahead of the replication forks is compromised and fork swivelling is thought to be the preferred method to relieve super helical tension at replication termination sites[51]. The consequence of fork rotation is the formation of DNA crosses between replicated sister chromatids behind the forks, or precatenanes, which become sister chromatid intertwines (SCIs), or full catenanes, upon replication completion. Importantly, because of the right-handed helicity of DNA[1], the fork swivelling produces right-handed crosses. Since TOP2α is likely to be the primary mitotic decatenase[13–15], the chiral preference for efficient right-handed braid resolution might enable the enzyme to be more effective at removing catenanes formed during DNA replication.

Our assay provides multiple outputs to measure the release of T-segment DNA in native conditions. Previously, type II topoisomerases have been shown to transport the T-segment through the cleaved G-segment in the absence of ATP hydrolysis[36,52,53]. In such ensemble assays, decatenation of singly catenated plasmids, in the presence of a non-hydrolyzed ATP analogue was achieved after washes with high ionic strength buffers (that is, containing 1 M NaCl)[36,52,53]. We did not observe DNA braid resolution when we substituted ATP for ATPγS or AMPPNP in our assays (Fig. 3a), despite the fact that TOP2α was able to bind to DNA braids (Fig. 3b). This could be due to three possible scenarios: 1. Failure to capture the T-segment, 2. Failure to transfer the T-segment through the DNA-gate and 3. Failure to release the T-segment from the C-gate. Given previous studies have demonstrated that ATP hydrolysis might not be necessary for transport of the T-segment through the cleaved G-segment, but is likely to be important for the release of the T-segment through the C-terminal gate, we speculate that scenario 3 is most likely. Our assays using etoposide are consistent with this. Previous work has suggested that etoposide acts before the release of the final ADP, allowing transit of the T-segment, but preventing re-ligation of DNA, thus resulting in decatenation of single catenated plasmids in assays where the reaction is followed by denaturing condition[54]. In our assays, etoposide is able to inhibit resolution of a single braid (Fig. 3a), suggesting it prevents release of the T-segment from the C-gate. Consistent with this, others have reported that etoposide can promote TOP2α DNA looping, potentially by preventing T DNA release[42]. Collectively this suggests that TOP2α may have a release checkpoint to ensure the G-segment is re-ligated before the release of the T-segment, as it has been previously demonstrated that the G-segment can be re-ligated while the T-segment is trapped in a covalently closed C-gate[35].

An advantage of our assay is that, as DNA is held under tension between two optically trapped beads, we can control how many DNA segments are bound by TOP2α. In ensemble experiments, DNA is readily available, making it difficult to distinguish whether one or two DNA segments are bound, either stably or transiently. Our data demonstrates for the first time that TOP2α can associate with a single dsDNA segment and diffuse along dsDNA, with the diffusion rate being influenced by the presence of ATP. Furthermore, as we can load TOP2α on to dsDNA, our assay also enables us to separate out the steps of DNA binding and braid junction localisation. We found that loading of TOP2α onto a single dsDNA, particularly in the presence of ATP, compromises the ability of the enzyme to resolve any newly formed DNA braids that it encounters (Fig. 3a). Furthermore, resolution was also impaired when DNA was rebraided after a resolution event, despite the fact that TOP2α is able to resolve multiple braids that are generated before incubation with TOP2α. Our results indicate that for efficient resolution TOP2α needs to load at or near the junctions, since this maximises the probability of capturing both DNAs, rather than diffusing along DNA to find braids. This is consistent with previous EM

studies which observed TOP2 preferentially binding at the crossing on two DNA junctions[40] and recent cryo-EM structures of the type II topoisomerase, gyrase, illustrating contacts are made between the DNA-gate and both the G- and T- DNA segments[55,56]. As TOP2α acts on chromatin and its DNA binding sites are too small for nucleosomes to fit through, DNA diffusion could be hindered by the presence of nucleosomes. Previous studies suggest that chromatin reduces TOP2α accessibility[57], although this may be overcome via chromatin remodelling, as TOP2α interacts with the BAF chromatin remodeller, promoting its recruitment[58]. We envisage that in substrates with multiple DNA braids, after each resolution, the enzyme has a high probability to capture the next two DNA segments for the next enzymatic cycle, prior to ATP closure of the N-terminal gate, thus explaining the processivity observed (Fig. 4b).

The observation that TOP2α pre-bound to dsDNA can in the absence of ATP resolve newly formed DNA braids, although with reduced efficiency (Fig. 2a), raises the interesting possibility that ATP binding by TOP2α molecules that have captured a single DNA might generate an intermediate where ATP-dependent closure of the N-terminal gate blocks capture of the second DNA, thus generating a situation where the enzyme stalls at a non-productive intermediate (closed state, Fig. 1b). A non-productive state may have to be removed by TOP2α removal pathways[59] or might explain the non-catalytic, structural roles of TOP2α that cell-based studies have demonstrated, like the observation that TOP2α degradation results in a chromosome decompaction phenotype not observed using enzymatic TOP2α inhibition[38].

Our results demonstrate that loading to, or close to, DNA crosses is a key determinant for rapid and successful decatenation by TOP2α. Our finding that cohesin complexes stably associate with braided junctions and have an inhibitory effect on TOP2α resolution of DNA braids (Fig. 5a and d) raises important insights into genome segregation. Interestingly, the presence of SCI in yeast chromosomes requires cohesin[20,21] and it has been well characterised that human centromeres are highly catenated regions whose decatenation by TOP2α occurs during anaphase[60], only after separase dependent removal of cohesin[15]. Moreover, recent work demonstrates that DNA replication forks push cohesin complexes to sites of replication termination, where forks converge[61]. Cohesin is maintained at these termination sites thus providing sister chromatid cohesion until mitosis[61]. Importantly, DNA termination regions are predicted to contain a high density of SCIs[2]. Our data suggests that cohesin stabilised at catenanes could prevent SCI resolution by topoisomerase II during interphase, and raises the possibility that cohesin enforces sister chromatid cohesion not only by holding DNA junctions between sister chromatids but also by preventing their premature resolution by TOP2α. We propose that cohesin inhibition of TOP2α could occur by restricting simultaneous capture of the two DNAs at catenation crosses, which we have shown to be an important requirement for efficient resolution of DNA braids by TOP2α.

In summary, we provide valuable insights into the possible regulation of TOP2α recognition of substrates held under forces, mimicking what could occur during the metaphase and anaphase stages of mitosis. We propose that rather than a mere passive presence within the nuclear milieu, randomly encountering and resolving DNA crossings, TOP2α function is likely to rely on a highly regulated orchestration of the enzyme access to specific DNA substrates. We anticipate that this regulation will include mechanisms that restrict access to regions where processing should be avoided, such as cohesion sites, as well as regulation that facilitates the presentation or capture of DNA crosses. Given the importance of human TOP2 enzymes as therapeutic targets and their increasingly recognized role as a potential source of genome instability, understanding how their activities are controlled and restricted to proper sites of action is an important question for the future.

## Methods

### Protein purification

Human TOP2α was encoded from the pLIB vector with a C-terminal 3C-ybbr-tev-strepII tag, Cohesin STAG1 tetramer from pBIG2ab[62] with a C-terminal 3C-His10 tag on SMC3 and a C-terminal 3C-ybbr-tev-strepII on STAG1 and NIBPL with a deletion of N-terminal 1162, an N-terminal MBP and C-terminal 3C-ybbr-tev-strepII tag from pLIB. All constructs were transposed into DH10EMBacY and purified Bacmid transfected into SF9 cells. After 72 hours, virus was harvested and further amplified in SF9 cells before being used for expression in either SF9 or HighFive cells for 72 hours. Cell pellets were resuspended in purification buffer (20 mM HEPES [pH 8], 300 mM KCl, 5 mM MgCl2, 1 mM DTT, 10% glycerol) supplemented with 1 Pierce protease inhibitor EDTA–free tablet (Thermo Scientific) per 50 mL and 25 U/mL of Benzonase (Sigma) and lysed with a dounce homogeniser followed by brief sonication. Lysate was cleared with centrifugation before being loaded on to a StrepTrap HP (GE), washed with purification buffer and eluted with purification buffer supplemented with 5 mM Desthiobiotin (Sigma). Protein containing fractions were pooled, diluted 2-fold with Buffer A (20 mM HEPES [pH 8], 5 mM MgCl2, 5% glycerol, 1 mM DTT), loaded on to HiTrap Heparin HP column (GE), washed with Buffer A with 250 mM NaCl, then eluted with a gradient up to 2 M NaCl. Finally, size exclusion chromatography was performed using purification buffer and a Superose 6 16/70 or increase 10/300 column.

Proteins were labelled using SFP transferase to attach HPLC purified dye conjugated CoA to the protein encoded ybbr tag[63]. TOP2α was labelled with Alexa555 while Cohesin labelled with ATTO647N. After protein labelling, complex was purified with size exclusion chromatography using a superose 6 increase 10/300 column. Labelling efficiency of TOP2α Alexa555 was initially ~30%, resulting in labelling of ~50% of dimers. TOP2α labelling efficiency was improved to ~70% by not cleaving the Strep-II tag, and performing a second Strep-II affinity column prior to size exclusion chromatography. The activity of the two TOP2α purifications however did not differ. The ~70% labelled TOP2α was used to collect force dependence data (Fig. 2a) and had a resolution probability ± standard error of 0.79 ± 0.03 at 5 pN, while ~30% TOP2α was used to collect all other datasets and had resolution probability of 0.83 ± 0.01 at 5 pN.

Cohesin STAG1-ATTO647N were ~30%. Human condensin I was purified and labelled as previously described 56. Labelling efficiency was ~90%.

### Mass Photometry

The molecular mass of recombinant complexes was confirmed with a Refeyn TwoMP mass photometer. Sample was applied to a Culture-WellTM gasket (GBL103250, Sigma-Aldrich) attached to a sample carrier slide (Refeyn). All samples were measured in 50 mM Tris pH 7.5, 150 mM NaCl, 2.5 mM MgCl$_2$ buffer using a field of view 512 × 138 pixels, collecting 6000 frames with a collection time of 60 s. The focal position and imaging conditions were set using a 12 μL buffer droplet and data was collected by adding 2 μL of sample, resulting in a final protein concentration of ~5 nM. All data were acquired with using the Refeyn AcquireMP software and analysed using the Refeyn DiscoverMP software. Masses were calibrated using the NativeMarkTM unstained protein standard (LC0725, Thermo Scientific) to generate a calibration curve.

### Decatenation Assays

Catenated kinetoplast DNA (kDNA, Inspiralis, K1002) (200 ng) was incubated with 80 nM of labelled or unlabelled TOP2α in 50 mM Tris pH 7.5, 125 mM potassium glutamate, 2.5 mM MgCl$_2$, 0.5 mg/mL BSA in the presence or absence of 1 mM ATP for 30 min at 37 degrees. Reaction was terminated with 3 μL of stop buffer (5% sarkosyl, 0.025% bromophenol blue, 50% glycerol) and incubated with 1.6 units of Proteinase K (NEB, P8107S) for 15 minutes at 37 degrees before

resolving on 1% agarose TAE gel stained with SybrSafe (Invitrogen, S33102). Decatenated products confirmed by comparison to control decatenated and linearised kDNA standards (Inspiralis, KD100 and KL100, respectively).

## ATPase assays

ATPase assays were performed with complexes of wild type or ATPase hydrolysis deficient Q-loop mutants of cohesin with 50 bp dsDNA, and with or without MBP-ΔN-NIPBL.

Assays were performed using the EnzChek Phosphate Assay Kit (Invitrogen) modified for a 96 well plate format[64]. Reactions contained 30 nM protein with 600 nM DNA. Final conditions included 1 mM ATP and a total salt concentration of 50 mM. Protein/DNA was pre-incubated in reaction mix without ATP for 15 min at room temperature before the reaction was started by addition of ATP immediately prior to putting it in the plate reader to track phosphate release. ATPase rate was determined using standard phosphate curve using linear fit of data in linear region.

## Single molecule braid resolution assays

Lambda DNA was biotinylation by end filling with Klenow DNA polymerase as previously described[65]. Optical tweezer assays were carried out on a Lumicks Q-trap system, with integrated microfluids and confocal fluorescence microscopy. Flow cell was cleaned with 5% bleach, water, 25 mM Thiosulfate, water, before being blocked with 0.5% pluronic and 2 mg/mL BSA in experimental buffer (50 mM Tris pH 7.5, 125 mM potassium glutamate, 2.5 mM MgCl2).

All assays were performed in experimental buffer with 0.5 mg/mL BSA except assays including 1% SDS, for which 125 mM potassium glutamate was substituted with 125 mM sodium glutamate, to prevent SDS precipitating with potassium. In all assays, beads were in channel 1, biotinylated lambda DNA in channel 2 and buffer in channel 3. Protein solutions of TOP2α or cohesin were used in channel 4 and 5 using a standard flow cell, and 1% SDS in channel 6 of 9 channel flow cell. TOP2α was used at 2 nM for all braid resolution assays, cohesin was used at 2 nM, with 4 nM NIPBL, and condensin I was used at 2 nM, with 1 mM ATP or ATPγS and 0.1 nM YOYO-1dye

Trapping laser was set at 100%, splitting ~67% across traps 1 and 2, to achieve equal trap stiffness of ~ 0.30 pN/nm. Force is detected on bead 1 and 4. Prior to experiments, beads are trapped and force calibration using bead power spectra is performed following manufacturer's instructions, in line with previously published quadruple optical tweezer methodsb[22–25,48]. All experiments were started by trapping one bead per trap, tethering DNA between bead pairs 1 and 2, and 3 and 4 and collecting force-distance data for each DNA, zeroing force at distances where force is expected to be minimal, that is, at distances less than 10 μm. DNA knots/braids were created using the Bluelake DNA knotting script written by Aafke van den Berg from Lumicks Github (https://github.com/lumicks/harbor Experiment automation/Creating DNA knots on the Q-Trap) in combination with manual control to prevent bead collision and overstretching of DNA. The script first moved bead 1 and 4, to a fixed distance of 9 μm apart, bead 1 and 2 a fixed distance of 12.5 μm apart, then moved the Z-height of bead 3 and 4 from focus at 2.8 μm to 1.2 μm, passing bead 3 under DNA1 held by bead 1 and 2, then to 4.4 μm, before passing bead 3 over DNA1 and returning to focus. After the script completed, beads were aligned into a rectangle with a force of ~5pN. This results in a right-handed twist, similar to that found in negative supercoiled DNA or produced by turning beads anti-clockwise. The script could also be looped multiple times, and in the reverse order, resulting in the opposite handedness. Force extension (FE) curves were collected after to confirm the braid had been introduced. FE data was collected to a maximum of 30 pN, as higher forces have been previous demonstrated to result in DNA interactions which cause deviations in expected FE curve[23].

In standard assays, beads were moved to a mean force of ~5 pN in the buffer channel prior to moving to protein channels. This was achieved by first moving bead 2, such that the force in bead 1 was 3-7 pN, then bead 3 was moved such that the force on bead 4 was 3-7 pN. Movement of bead 3 could increase force of bead 1, but average force across bead 1 and 4 was maintained at 5 ± 5 pN. In higher force experiments, beads were similarly moved to forces fixed forces at 5, 15, 25, 30 and 45 pN, by maintaining a fixed distance of 7.8 ± 0.3 μm between bead 1 and 4, and varying the distances between bead 1 and 2, and 3 and 4 to achieve similar forces on bead 1 and 4. To determine mean and errors in force, the first 670 ms of force on bead 1 and 4 were averaged.

Two-dimensional scan data was collected to visually determine if TOP2α was bound, if the braided substrate had resolved and if the DNA molecules were intact. The structure and integrity of the DNA substrate could also be extrapolated from force data. Images were acquired with a pixel size of 100 nm x 100 nm, a pixel time of 0.1 ms, with 1% green laser (λex = 532 nm at 5 mW), and 2% red and blue laser (λex = 638 and 488 nm at 0.9 and 0.2 mW, respectively). Substrates were imaged upon moving into the protein, then at later time points to reduce laser induced DNA damage. A 5-minute analysis window was selected, and while longer time points were collected in some cases, no additional resolution events occurred using a larger analysis window.

Two metrics were used to confirm resolution: force and imaging. Once resolved, the force as measured on bead 1 and 4 dropped to essentially 0 pN. The loss of force often resulted in loss of the DNA from the imaging plane. To confirm DNA was still present, DNA force extension data was collected, such that an increase in force could confirm the presence of DNA, as well as move the DNA back into the imaging plane, for visual confirmation of DNA between the beads.

## General data analysis

Data was analysed in Lakeview (LUMICKS), Fiji and python using pylake. Force vs time curves were plotted with a down-sampling rate of 1000 Hz. Kymographs of condensin and cohesin at braid junction were generated from scan images using KymographBuilder 2.1.1 in Fiji using a jointed 10-pixel line.

## Rate estimation

Rates of TOP2α decatenation were estimated from survival probability, where

$$Pr_{Survival}(t, \lambda) = e^{-\lambda t}$$

For time, t, in seconds, and rate, λ, events per second. Rates were converted into time per event, for comparison with previously published experiments.

## Geometry analysis

A python script was used to estimate the crossing angle from the scan data (Supplementary Fig. 3). This analysis uses the centre position of the beads determined through bead tracking implemented in the Bluelake software. It assumes that the length of DNA 1 and 2 is equal and uses the SLSQP minimisation implemented by the scipy python package to find the DNA crossing point X, to minimise:

$$\min \left[ \left( \left| \overrightarrow{B1cX} \right| + \left| \overrightarrow{B2cX} \right| \right) - \left( \left| \overrightarrow{B3cX} \right| + \left| \overrightarrow{B4cX} \right| \right) \right]$$

Where B1cX is the vector defined by points B1c and X, B2cX is the vector defined by points B2c and X, etc.

Once X is found for a given bead positions, crossing angle, α, can be determined using linear algebra, that is, by finding in inverse cosine of the dot product divided by the multiplied modulus's of the vector

B1cX and B4cX.

$$\alpha = \cos^{-1}\left(\frac{\overrightarrow{B1cX} \cdot \overrightarrow{B4cX}}{|\overrightarrow{B1cX}||\overrightarrow{B4cX}|}\right)$$

This approach allowed estimation of crossing angle and geometry from bead positions in the absence of DNA dyes. Crossing angle estimates were compared to those measured directly from scans of YOYO stained DNA, where DNA was visible for at least three frames (Supplementary Fig. 3b). The average ± standard deviation measured crossing angle for directly measured data was $68.9 \pm 0.7°$, with a corresponding estimate of $67.0 \pm 2°$, with no significant difference across the different methods for determining crossing angle.

The script also provides an estimate for the braided DNA length:

$$\text{DNA length} = \left|\overrightarrow{B1cX}\right| + \left|\overrightarrow{B2cX}\right| - 2 * \text{bead radius}$$

The presence of braids can be confirmed by collecting FE curves, where plotting the force versus the distance between bead 1-2 or 3-4 results in DNA appearing shorter than naked, non-braided DNA (Supplementary Fig 3c, red vs pink curves). However, the braided DNA distance estimate shifts the braided FE curve towards that of the naked DNA (Supplementary Fig. 3c, green vs. pink curves).

Crossing angles were consistent across experiments, except for data collected at 45 pN, which exhibited a slight decrease (Fig. 2d and Supplementary Fig. 3d and e).

### Fluorescence Intensity analysis
The fluorescent intensity of TOP2α-Alexa555 localised to the junction of a DNA braid was determined from a 2D confocal scan by fitting the fluorescent spot using lmfit (version 1.3.2) to a symmetric 2D Gaussian equation:

$$g(x,y) = \text{height} * \exp\left(-\left(\frac{(x-x_0)^2 + (y-y_0)^2}{2\sigma^2}\right)\right)$$

with peak centre $x_0, y_0$ and variance $\sigma^2$.

Green pixel intensity from kymographs was extracted using scripts provided by Benjamin Ambrose and steps fit with Stepfinder (https://github.com/tobiasjj/stepfinder) using an expected step size of 1, step size threshold of 1.

### Statistical analysis
Statistical analysis was performed utilizing Bayesian inference, where the catenation event was modelled as a Bernoulli trial with a decatenation being defined as a success. Since the likelihood for this was a binomial distribution, the probability of decatenation could be estimated as a beta distribution, Beta (α, β) Using the beta uniform distribution, Beta (1,1) as the conjugate prior, the parameters for the above distribution could be calculated as $\alpha = k+1$ and $\beta = n\text{-}k+1$, where n is the number of trial and k is the number of resolution eventsIn line with this analysis, the probability of resolution would have

Mean = $\alpha/(\alpha + \beta)$, and

$$SD = \sqrt{\alpha\beta/((\alpha+\beta)^2(\alpha+\beta+1)}$$

Data in figures and text is reported with mean ± standard error.

### Diffusion analysis
Kymographs with diffusing traces of Alexa555 labelled TOP2α were analysed using a custom single-particle tracking algorithm[39] utilizing the Pylake package and standard Numpy, Matplotlib, SciPy, and PeakUtils libraries The Mean Square Displacement (MSD) of each tracked molecule was calculated. The MSD analysis was done by fitting a 1D Gaussian function to the signal intensity over a 3-timeframe moving window.

For each diffusing molecule analysed, the following equation was used:

$$MSD(n, N) = \sum_{i=1}^{N-n} \frac{(X_{i+n} - X_i)^2}{N-n} = D\tau + b,$$

where $N$ = total number of timeframes in the kymograph, $n$ = number of frames within a moving window of time ($\tau$) from which the square displacement was calculated (from 1 to $N-1$), and $X$ = position of the molecule along the DNA. The slope of the linear fit model ($D\tau + b$) is the 1D diffusion coefficient ($D$). The MSD was fit between $0.15 < \tau < 1\,\text{s}$ (3-10 lag times). Each point in the MSD plot represents a line in the scan, and line times were 60.5 ms for TOP2α kymos and 57.9 ms for cohesin kymos, with a pixel time of 0.2 ms.

Igor Pro 8 (Wavemetrics) was used to plot and fit the Gaussian distribution of log(D) for the diffusion coefficients calculated ± ATP using the following function:

$$f(D) = y + A_1^{-\left(\frac{D-D_1}{w_1}\right)^2} + A_2^{-\left(\frac{D-D_2}{w_2}\right)^2}$$

where is the baseline and are the amplitud, mean, and width of each peak. The no ATP condition was fit to a single Gaussian model and the with ATP condition was fit to a double Gaussian model.

### Reporting summary
Further information on research design is available in the Nature Portfolio Reporting Summary linked to this article.

## Data availability
All raw data supporting the findings of this study have been deposited in the Zenodo repository and are publicly available at [https://doi.org/10.5281/zenodo.15704511]. Source data are provided with this paper.

## Code availability
The scripts used for geometry analysis is available at Github (https://github.com/Cutts-Lab/Qtrap_analysis) and has been deposited in the Zenodo repository and are publicly available at [https://doi.org/10.5281/zenodo.15704511].

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

## Acknowledgements

The Aragon and Rueda labs are supported by the Medical Research Council [UKRI MC-A652-5PY00 and UIKRI MC-A658-5TY10, respectively]. E. E. Cutts was awarded a Dean's award seed grant by the Institute of Cancer Research which was used to purchase the human topoisomerase alpha expression construct.

## Author contributions

E.E.C. and L.A. designed experiments. E.E.C. cloned, expressed, purified all proteins. E.E.C. performed experiments and analysed the data. S.S. contributed to collection of fixed force data and performed experiment with multiple left-handed braids and diffusion analysis. P. G. performed fluorescence intensity analysis, assisted by scripts from B. A. G.L. M. F. designed the cohesin bridging assay. D. S. R. designed chirality assays. E.E.C. and L.A. wrote the manuscript. L.A., D.S.R., and E.E.C. reviewed the manuscript. All authors contributed to editing and provided additional text for the manuscript.

## Competing interests

The authors declare no competing interests.
