## [Transparent Peer Review file · Nature Communications]

Substrate accessibility regulation of human TopIIa decatenation by cohesin

Corresponding Author: Professor Luís Aragón

Version 0:

Reviewer comments:

Reviewer #1

(Remarks to the Author)

In this study, Erin E. Cutts and colleagues investigate the mechanism of TOP2a decatenation using braids of lambda DNA generated using quadruple-trap optical tweezers. They observe that TOP2a is able to untangle both single and multiple DNA braids, and that this is more efficient for right-handed braids than left-handed braids. Furthermore, they find that the untangling ability of TOP2a is inhibited by loading onto DNA in absence of a second DNA strand, as well as the presence of etoposide, non-hydrolysable ATP, and the presence of cohesin. Although the manuscript has the potential to provide valuable single-molecule insights into the function of TOP2a, there are some important concerns that should be addressed prior to its publication.

major comments

Fluorescence

I have some concerns about the quality and interpretation of the fluorescence micrographs. First of all, the signal of all used dyes appears to be very dim, making it difficult to see exactly what is happening in each image. Looking at the methods section, it seems that the authors used low fluorescence excitation laser powers. The authors should convincingly show that they have the single molecule resolution they claim to have (for example in line 275), using a study of photobleaching traces for all used fluorophores. If this analysis shows that single molecule resolution was not obtained, the experiments should be repeated under optimized conditions, or conclusions should be adapted accordingly. If/when single molecule resolution is obtained, it would be very insightful to convert the fluorescence intensity to the number of bound molecules, so conclusions could be made about the number of molecules responsible for certain events etc.

The labelling efficiency of TOP2a and especially cohesin is quite low. The authors should consider and discuss how this affects their data interpretation and conclusions, for example in line 275: 'it allows us to see the outcome of every TOP2A that binds'.

On several occasions the translocation of fluorescently labelled TOP2a is interpreted as diffusion. However, there isn't currently any analysis of the tracks of fluorescent particles to support this interpretation.

Force

Both from cited sources and new data provided by the authors, it seems apparent that force greatly impacts TOP2a activity. However, the force data and its description presented in this work does not give sufficient information to truly appreciate the role of force in TOP2a-mediated braid resolution, and to rule it out as a confounding factor between experimental conditions. It is currently unclear in most cases from which bead the mentioned force is measured, and only an estimate of the applied force is given. The exact same force should be applied to the same bead for each experimental condition to obtain reliable results. Especially for the comparison to 'high force' of >30 pN the lack of precision is very problematic. The force should be well calibrated and kept constant during measurements. For force calibration of quadruple-trap optical tweezers, Brouwer et al., 2016, Optical Tweezers, Methods in Molecular Biology (https://doi.org/10.1007/978-1-4939-6421-5_10) might be a good starting point. At minimum, a (supplementary) figure depicting the actual applied tension versus the resolution probability should be provided for each condition. Additionally, it would be better to give forces in figures as absolute instead of relative values.

Timescales

For all of the investigated conditions, the timescale at which the braid resolution does (or does not) take place is quite relevant. Currently, there is no mention of these timescales. In my opinion, it should at least be mentioned for how long each measurement was conducted and after how much time the resolution took place. Additionally, timestamps should be included instead of frame numbers for all fluorescence images.

Statistics

Throughout the manuscript, there are a few statements indicating the frequency of a particular event/finding, without mention of the number of events that support these claims. The N of measurements should be included in the following instances:

- Line 164: 'in many cases'
- Line 196: 'in most cases'
- Line 221: 'TOP2a remained bound to the DNA after the resolution events'
- Line 236: 'kymographs show that...' No N given for number of kymographs or number of tracked proteins
- Line 320: 'kymographs show that...' No N given for number of kymographs or number of tracked proteins

Some conclusions are supported only by slim statistics ($N < 10$). Especially when statements are made that events never occur, the authors should take care not to overinterpret their data.

General

The authors should look more careful at the available literature on quadruple-trap optical tweezers experiments. This has been around for ± 15 years and quite some of the issues encountered here are solved and/or discussed before. They can thus provide answers and should be properly cited.

minor comments

Line 35: here >20 pN is mentioned, whereas later in the text >30 pN is mentioned. Which one is correct?

Line 146: it would be useful here to mention that the activity of TOP2a was tested using decatenation assays.

Line 150: 'incubated the DNA with protein': What was the incubation time?

Line 236: the reasoning behind the switch from Alexa555 to Cy3B should be given. Also, the labelling efficiency of TOP2a-CyB should be given in the methods section.

Line 299: it is not immediately clear to me how this conclusion is in line with the data presented in this section.

Line 468: 'Pulling on DNA1 results in no force in DNA2'. Looking at figure 1d, the force in DNA2 in fact increases from 0 up to ~ 4 pN. How is this drift in force explained?

Line 599: what was the reason behind the switch from potassium glutamate to sodium glutamate?

Line 629: when describing the fluorescence laser settings, both the wavelength and power (in mW) should be mentioned. Furthermore, details about the confocal imaging, such as pixel size and pixel dwell time should be given.

Figure 1b: the cartoon indicates overstretching, but no overstretching plateau is seen in the graph, as the graph only extends to 60 pN. Additionally, given the specifics of the DNA construct, overstretching in the cartoon should start from the ends, not the center.

Figure 1d: it is unclear whether the distance from bead 1-2 or bead 3-4 is plotted. It also could be misleading to the reader that the scaling on the y axis is quite different between both panels. Lastly, the symbols in the lower panel do not match the figure legend.

Figure 2a: from this panel it is unclear that YOYO was present in the ATyS condition.

Figure 2d,e: it is not indicated what the dotted line in normalized fluorescence intensity plot represents. Additionally, it now appears that there is a time correlation between the fluorescence frames and the force plot, is this really the case?

Figure 2e: in frame 2 there appears to be a DNA horizontally between the lower beads. Could it be that the braid was resolved during image acquisition?

Figure 2e: if the braids are resolved in several steps instead of simultaneously, for each resolution only a miniscule amount of DNA would be released. How can this be unified with the significant force drops observed here? To help answer this it would again be useful to look at the absolute instead of relative forces.

Figure 2f: it would be helpful for the interpretation to state the force at which the DNA was held for these kymographs.

Figure 4a: the DNA appears to be curved in this example. How were the kymographs generated from a curved molecule?

For all figure panels containing fluorescence images, a scale bar based on the used pixel size should be included. Using the trapped beads as a scale is inaccurate and insufficient.

(Remarks to the Author)

The manuscript by Cutts et al investigated human Top2 α decatenation activity and cohesin dependent regulation of Top2 α at the single molecule level in real time. The authors presented some interesting observations that could be of interest to the topoisomerase community. However, the measurements and the interpretation are somewhat flawed without time information. The authors modeled all the measurements as Bernoulli trials, treating each measurement as representing essentially a binary probability of success or failure. The decatenation activity by Top2 α , even with a single-braid, is difficult to characterize as a coin flipping event modelled as Bernoulli trial mentioned in the method without defining the time over which the trial is performed, which must be constant for each measurement. As such, I would recommend the authors measure the waiting times from Top2 α localization at the DNA cross for individual decatenation events in order to characterize Top2 α activities under different conditions. Regardless of “not heterogeneous substrate”, the decatenation probability estimated based on the small number of events would be highly biased compared to the ensemble measurements using heterogeneous substrates. A similar issue also caught my attention. The authors compared the decatenation activities among a few prebound enzymes before braiding and DNA braid with 1-2 nM of enzyme. Enzyme target finding process inherently takes time through a mixture of 3D and 1D diffusion (Halford et al; NAR 2004). Naturally, it takes longer to observe the activity with lower concentration of enzyme than higher concentration as previously tested at the single molecule level (Yogo et al; PLOS One, 2012) where no decatenation activity was observed at 0.1 pM for 300 s while <1s at 1 nM in the presence of 30 braids. Thus, reducing number of enzymes from 2 nM to a few should significantly increase the waiting time to observe decatenation activity, which makes unfair to conclude that preloaded enzymes become inactive.

The data shown in (Fig. 4) presented directly reveals that Cohesin can physically hinder Top2 α from accessing DNA crossing providing an insight on the regulatory role of cohesin in the separation of two sister chromatids by Top2 α . Taking advantage of the technique, it would be interesting to test the other possible Cohesin roles (Piskadlo et al; Int J Mol Sci 2017) – promoting Top2 α catenation activity by increasing proximity of two DNA. For example, the authors can test catenation activity by Top2 α using DNA configuration shown in Fig. 4f, panel 3.

The manuscript presents interesting observations and potentially important insights on Top2 α and cohesin interactions. However, there is a caveat in the measurements and the interpretation. Thus, without a major revision, I would not recommend it for publication in Nature Communications.

I would recommend the authors to include the following information for better understanding of the study:

1. Cut-off time before considering no decatenation, and more generally for every measurement. As it stands there is not a single-mention of the durations of individual measurements. Without this information, and unless the durations were identical, the results are extremely difficult to interpret. If the waiting time for each measurement was constant, then the authors have to clearly define the measurement approach, including how the durations were chosen, and why the more conventional and informative kinetic or dynamic measurements were not performed. Dependent on how long the authors waited before considering no decantation, decatenation probability would change and hence, measuring the rates with repeated frequency are recommended. Furthermore, reducing kinetic measurements to probabilities makes comparisons to previous measurements, or interpretations of any implied kinetics, impossible. The authors should therefore provide a comparison between their observed probabilities and the probabilities that would be expected based on prior measurements of Top2 α activity, which will once again require well-defined and constant waiting time intervals.

2. The average waiting times for decatenation events tested in the study.

3. The frequency of Top2 α localization at crossed junctions during decatenation measurements listed in Fig.2a. For example, If N=10, on how many occasions did the authors observe Top2 α localization at cross junction?

Page 4; line 136-137: Could the authors clarify which characteristics of the braids generated by four beads would be similar to that of the catenates of two plasmids? Based on the simple geometric comparison, a crossing by the arrangement of four beads in this article should have larger crossing angles than those of two plasmids, which could affect efficacy of Top2 α decatenation activity.

Related to making a right-handed crossing, the description of the figure caption 1 c) “bead 3 and 4 are moved up in Z” doesn’t seem correct. It should be “lowering bead 3 and 4 in Z” relative to the plane of bead 1 and 2 in order to place DNA 2 beneath DNA1.

Fig. 1d DNA 2, triangle should be rectangular for DNA braid trace. And it would be clearer if different colors are used to distinguish Naked, braid, and DNA after TOP2A instead of different shades of red.

I do not understand DNA 2 force extension curve. Shouldn’t the force and extension for DNA 2 remain constant when it was not braided or after decatenation by Top2 α when only DNA 1 was under a stretching force?

Fig. 1e and g. Could the authors explain why more Top2 α fluorescent dots appeared on DNA after DNA unbraiding?

Page 5; line 169- 183: The authors tested the idea that ATP hydrolysis is required for releasing of transfer DNA from C gate and concluded that the lack of braid resolution with ATPYS by Top2 α proved this idea. I think that the authors overinterpreted the result without obtaining direct evidence that the transfer strand is captured by the enzyme under these conditions. The lack of resolution could be that Top2 α bound at the crossing without decatenation. The only way to confirm capture of the T-segment in the C-terminal cavity is by either establishing that the DNA molecules are topologically linked by after reversing the crossing or by removing protein to see if the braid was resolved. T-segment capture in the C terminal gate in the presence of AMPPNP was directly observed by Martínez-García (NAR 2013). Additionally, there is some evidence showing that ATPYS is slowly hydrolysable, so AMPPNP would be a better ATP analogue for this assay.

Page 6; line 185-201: The authors investigated Top2 α resolution of multiple braid.

Multiple braids resolutions by different type IIA topoisomerases have been extensively investigated in previous studies at the single molecule level by Yogo et al, Charvin et al (PNAS 2003) in addition to ref 32 that elucidated the processivity of topo IIA on different braid geometry, chirality, and tension dependence. In this manuscript, the authors replicated the previous works with less rigor. 4 braids is not enough to investigate processivity as it corresponds to two uncrossing cycles for topo

IIA. Also, given the relatively high enzyme concentration with one substrate in contrast to the previous works where the enzyme concentrations were as low as a few pM, one should expect multiple enzymes would potentially bind and resolve DNA braids. Perhaps, taking an advantage of fluorescently labeled Top2 α , it would be informative to measure an average processivity of a single Top2 α by applying high number of braids, which was difficult in other studies without fluorescent labeled enzymes. Additionally, it would be interesting to see where Top2 α prefers to bind along DNA braids. It has been suggested for topo IV to prefer at the end of braids while yeast topoll appeared to have no particular preference (Charvin et al; BJ 2005).

Page 7; line 204-219: The authors investigated the inhibition of Top2 α by etoposide. However, considering the mechanism of etoposide binding, it is unclear why Top2 α could not resolve 4 braids that only takes two catalytic cycles. As shown in the previous study (ref. 31), etoposide dependent blocking of catalytic cycle of topo II was transient and only slightly slowed down the catalytic rate with increasing etoposide concentration. At 50 μ M etoposide, reference 31 showed a minor reduction in the rate for human Top2 α compared to no etoposide suggesting etoposide dependent inhibition is transient, not indefinite. Could the authors explain what would cause such discrepancy?

Page 7; line 220-264: Based on numerous appearances of Top2 α fluorescent dots, it is possible that the reason for the failure of decatenation by bound proteins were not the same "active" enzymes that could decatenate shown in fig. 1e and 1g. Were those bound proteins able to diffuse along DNA? I think that the detection of diffusing Top2 α shown in fig. 2f were taken in 1-2 nM enzyme concentration. For rebraided (N=10) and preloaded (N=9) experiments, how often did the authors observe enzyme localization at the crossing and enzyme diffusion?

The authors suggested that the inability of preloaded Top2 α enzymes in the presence of ATP to decatenate was either due to wrong binding geometry or locked N-gate to capture of the second DNA. However, for wrong binding geometry argument, if Top2 α can diffuse, it could reorient itself in principle but it would take time to encounter crossing before diffusing off from DNA. For the ATP-dependent locked N-gate suggestion, I could argue that ATP hydrolysis will reset the blockage as Top2 α is able to undergo ATP hydrolysis with linear DNA. If ATP hydrolysis is hindered and N-gate is locked for some time, there would likely be differences in the enzyme diffusion rates with and without ATP as the enzyme would remain bound and less diffusive. In principle, the authors could measure them. I think it would be interesting to see.

Page 8-9: The authors investigated if Cohesin could either block DNA separation after decatenation by Top2 α or prevent Top2 α decatenation activity all together. For measurements, the authors tried two different monovalent conditions. Could the authors explain why two different monovalent salt conditions were tested (KGluc and NaGluc)?

Cohesin can load on to a linear DNA easily but it could be inefficient to load onto and encircle a braid (Fig. 4e) as it has to capture both DNA within a pore. Alternative possibility would be dimerization of two Cohesins at the crossing. Could the authors distinguish these two possibilities?

Minor point: "and" is missing in the sentence "direct interaction between Top2 α the BAF.."

Reviewer #3

(Remarks to the Author)

In the manuscript by Cutts et al., the authors use optical tweezers to visualize how human TOP2alpha enzyme enables passage of one DNA through another and in the case of one and four "braids". Interestingly, authors show that in their conditions, where DNA is torsionally relaxed, TOP2 can resolve concatenations under loads of up to 30 pN. Finally, authors visualize how binding of cohesin to a DNA intersection prevents TOP2 from resolving it.

Major points

1. There seem to be several problems with an assay in which TOP2 resolves multiple braids:

- TOP2 is expected to resolve multiple braids processively. However, authors report that in most cases multiple braids were resolved in a single step. Is this the result of a single TOP2 action? If yes, why doesn't a trace consist of multiple consecutive steps corresponding to resolutions of consecutive braids? Is the resolution of the optical trap sufficient to detect steps associated with decatenation of individual braids? If not, it should be straightforward to create more braids and visualize how multiple braids are resolved continuously.
- Does the same TOP2 molecule continue resolving consecutive braids after it binds the intersection, or do multiple TOP2 molecules exchange to resolve concatenation with four braids? Does the fluorescent signal come from a single or multiple TOP2 molecules?
- Counterintuitively, several steps are indeed detected when brighter TOP2 signals are present (presumably due to multiple TOP2 molecules). How large are these steps and what is the distribution of their sizes? Are sizes consistent with a single or multiple braids being resolved? (Bottom figure 2e does not have numbers on y-axis and this information does not seem to be anywhere in the text). There is a contradiction in the data, which currently suggest that that multiple TOP2 resolve braids in multiple steps while at the same time one TOP2 resolves all braids in one step.

2. It is an interesting observation that TOP2 can diffuse along the DNA, and it is surprising that diffusing molecules are much less efficient in resolving DNA braids. Authors conclusion that "TOP2 sits on DNA with geometry incompatible with the capture of the second DNA" is possible, but without any quantification or any other experiments to support this, a simpler explanation seems more likely. Decatenation efficiency depends on the on-rate with which TOP2 molecules bind DNA intersection. With 2 nM TOP2 in solution the on rate is likely much larger (given the vast number of molecules around available for binding) than the rate at which a single TOP2 diffusing on DNA can run into the intersection within the

reasonable time of the experiment. This may also explain slightly higher efficiency of decatenation without ATP: although there is no quantification, fig 2f seems to suggest that there are more TOP2 molecules bound to DNA in the absence of ATP, and they diffuse more rapidly. This explains why single TOP2 molecules in the absence of ATP are more likely to run into the intersection and decatenate it, while there is more restricted diffusion in the presence of ATP, which makes these events extremely rare and probably unobservable within the timeframe of the authors' experiment.

3. More general comment related to the one above is that quantification of the binding/unbinding rates of TOP2 and its diffusion coefficients could help to distinguish between possible interpretations of data.

4. Experiments in which authors show that cohesin blocks TOP2 activity support earlier studies proposing that cohesin protects centromeric regions from decatenations. The presented method of visualization does have the potential to reveal interesting molecular details, but they seem to be missing at this point.

a. Do authors suggest that the inhibition is purely mechanical? If yes, would any bulky protein prevent TOP2 decatenation, or does cohesin have any unique activity, which allows it to perform this specific function?

b. It is interesting that cohesin that can diffuse on one DNA and becomes trapped at the DNA intersection. Do all cohesins do that, or is that just that one example? If cohesin has ability to track and block DNA intersections wherever they go, that would be an interesting mechanism for TOP2 inactivation, but I am not sure if authors' results are sufficient to claim this or even if that's what they aimed to do. Some quantitative data would be needed to support or rule out this model.

c. What is the topological state of cohesin at the intersection in these experiments? Does its ability to block decatenation by TOP2 depend on whether it is topologically or not topologically bound?

d. Does the cohesin inhibition depend on tension?

Minor

1. Fig.1. What are the on and off rates of TOP2 on single DNA versus DNA intersections?

2. Fig. 1f. doesn't have numbers on y-axis. What are the steps?

3. Representation on Fig2a is misleading (Same for Fig. 4b). First bar is 8/8, which should be one. Why does it correspond to 0.9 in the left axis? Same for 0/9, which should be zero, but it is ~0.1 on the axis.

4. It is unclear how error bars were calculated. SD for a binomial (or beta) distribution with the mean equal to zero is also zero. What do then error bars on 0/14, 0/9, 0/10 and 0/9 represent? What is the point of using error bars anyway if exact numbers are given? It may be more useful to have a measure of statistical significance of the difference between the experiments.

5. Figure 2d and 2e middle graphs. What should this fluorescence intensity be compared to? What is the expected signal from a single fluorophore?

6. Figure 2d and 2e bottom graphs. Units on y-axes are unclear. How big is one step? What is the distribution?

7. Fig. 2f, do TOP2 on and off DNA binding rates and diffusion rates depend on its ATP state?

8. Assuming K_d of etoposide is in the 1 μM range, concentration of the drug of 50 μM and TOP2 is 2nM, the ratio between the drug bound and unbound forms of TOP2 should be ~ 50 . Authors report that efficiency of DNA cleavage for one braid drops ~ 2 times when 98% of TOP2 molecules are presumably inhibited. Is this a contradiction?

9. What counts as a number of trials in quantifying decatenation events (Fig. 2a)? Presumably these are the events in which TOP2 was visibly bound at the intersection. How long does TOP2 need to stay at the intersection to be counted as a single trial? How far can it bind? What was the off rate for TOP2 in unsuccessful events?

Version 1:

Reviewer comments:

Reviewer #1

(Remarks to the Author)

General remarks:

Although the authors have managed to improve the manuscript in some important ways, in my opinion there are still remaining structural flaws (see comments below). More generally, the manuscript feels kind of rushed and should spend more attention to details. Therefore, in my opinion, the manuscript is not yet ready for publication in Nature Communications.

Below are specific comments on some of the answers given:

Regarding the answer on our previous concerns on Fluorescence:

It is difficult to compare intensities in figure Supplementary Figure 3c and Supplementary Figure 4d, as intensities in Supplementary Figure 3c were normalized. In the current state, I don't see added value in Supplementary Figure 3c. It is unclear if the lowest normalized intensity in Supplementary Figure 3c corresponds to single molecules, for example. Line 182-184: "Although we observed TOP2 α localisation at the DNA junction, no resolution events were observed when ATP γ S was present (N=14, Fig. 2a, Sup Fig. 3b and c)." Supplementary Figure 3c does give any information to support this claim. Figure S4d is not mentioned in the main text where it is relevant in this context.

The diffusion analysis (especially the figures) is not well explained appears incorrect. Some examples: In Supplementary Figure 4c there are a lot of data points within 1 second, but this does not seem to make sense given the line scan time. In Figure 2g one of the calculated values for D is negative.

See King et al. 2018 Nucleic Acids Research for an example of diffusion analysis on optical tweezer data.

Diffusion analysis should also be performed for cohesin, as the authors mention this molecule also diffuses along the DNA (Fig. 4a).

Regarding the answer on our previous concerns on Force:

5 \pm 5 pN is a very strange force range to give.

The authors still do not show any improvement for the high force (>30 pN) case, where their argumentation about noise during force clamp will not hold. The main problem here is that only a lower threshold is given. With this definition, experiments in this condition could have been performed over a force range of tens of piconewtons, making the data impossible to interpret. Once again, the force should be as similar as possible between measurements of the same experimental group. In a force clamp mode where the force is set by the user, this really should not be a problem.

Our earlier reviewer comment:

At minimum, a (supplementary) figure depicting the actual applied tension versus the resolution probability should be provided for each condition. Additionally, it would be better to give forces in figures as absolute instead of relative values.

your reply:

As explained above, due to the limitations of the Q-trap system in obtaining accurate tension measurement at the braid when TOP2a decatenation takes place (due to the necessary movement of the traps after calibration across different channels), we cannot provide exact force values for each braid. Instead, we state the force of resolution as an average of 5 \pm 5 pN for braids tested when the DNAs are taut (in Figures 2a, 3b and 4d) and >30pN for our experiments when DNAs are held at higher forces (Figure 3b; >30pN).

Our answer to this:

See comment above. No improvement is given for the high force case. This remains an important point to improve.

Our earlier reviewer comment:

1.12 Line 629: when describing the fluorescence laser settings, both the wavelength and power (in mW) should be mentioned. Furthermore, details about the confocal imaging, such as pixel size and pixel dwell time should be given.

Our reply:

We agree with this reviewer. This information has been added to Materials and Methods on page 16 (Wavelengths are 488, 532 and 638 nm; Power is 0.2, 5 and 0.9 μ W, respectively; Pixel size is 100 nm² and pixel time is 0.1 ms).

Our answer to this:

Instead of μ W the authors probably mean mW.

Pixel size is likely 1000 nm², corresponding to a pixel of 100 nm by 100 nm.

Reviewer #2

(Remarks to the Author)

As additional information has emerged in the rebuttal provided by the authors response to the reviewer's critiques, the overall strength of the manuscript has decreased and I am less certain about the results, the conclusions, and the suitability of the manuscript for publication in Nature Communications. For example, the stated protein labelling efficiency dropped from 70% in the original submission to 30% in the revision. Furthermore, the relatively slow scan rate (0.1ms/100nm²pixel) was only reported in the revised manuscript along with the unusually large uncertainty in the force (5 \pm 5pN). Overall this new information and the discrepancies between the initial submission and the revised submission have diminished my confidence in the results and conclusions and therefore my enthusiasm for the work. This critical information should have been included in the initial submission, which indicates a lack of adequate rigor.

The revisions and rebuttal also reveal that the vast majority of the results are simply recapitulations of previous measurements, albeit with a new technique that generally appears to produce inferior data. It would be reasonable to include one figure describing "proof-of-principle" measurements reproducing previous studies for the purpose of showing the robustness of the enzyme and experimental approach. However, most of the TOP2 α decatenation results do not provide any new information and rather reconfirm previously established results "qualitatively" or produced different outcomes compared

to previous studies for which the authors did not provide a reasonable explanation for the discrepancies. Moreover, the reliance on time-limited trials to measure enzyme activity is a step back from the real-time kinetic measurements that are now commonplace in current single-molecule measurements of topoisomerase activity (e.g. references ref. 35, 37 and 38). Thus the majority of the TOP2 α decatenation results reproduce prior measurements whereas the potentially new results (cohesion-dependent TOP2 α inhibition) are not performed with sufficient rigor or precision to justify the novel conclusions.

The results relating to cohesin dependent TOP2 α regulation could be of interest, but the high force (5 pN or higher), large force uncertainty (100%), and ambiguous, poorly controlled, crossing geometry, which has been shown to influence TOP2 α activity, with relatively short measurement duration trials (5 min) render the main conclusion circumstantial or ambiguous. Overall, I found that the revision would require significant improvements in data quality, control of experimental variables (force, crossing geometry), and statistics, not to mention kinetic measurements of enzyme activity. Additionally, the authors' responses to my major concerns were often unsatisfactory and even contradictory. Regrettably I cannot recommend the manuscript to be published in Nature Communications.

Specific points

The labeling efficiency information confusion:

In the previous version, "Labelling efficiency of TOP2 α Alexa555 and cohesin STAG1-ATTO647N were ~30%, and improved to ~70% for TOP2 α by enriching using the Strep tag after labelling prior to gel filtration."

In the current revision, "Labelling efficiency of TOP2 α Alexa555 and cohesin STAG1-ATTO647N were ~30%."

70% to 30% labeling efficiency would change the fraction of TOP2 α dimers containing at least one labeled monomer from 90% to 50%, thus significantly reducing the ability to directly observe TOP2 α dimers bound to the DNA, and increasing the likelihood of "dark" proteins acting on the DNA.

Due to the low labeling efficiency and low temporal and spatial resolution, the detected TOP2 α at the crossing might not be the one that is active. This also raises questions related to the coupling between TOP2 α localization at the crossing and decatenation activity.

In line with this, I previously asked why more enzymes seemed to appear after braiding resolution (fig. 1e and g). However, the authors reply (below) does not fully explain:

"TOP2 α can still bind to dsDNA after braiding because it is still present in the channel. We have now clarified this in the main text (Page 5)."

Before resolution (47s incubation in the protein channel; fig. 1e), there was one small bright dot indicated by the arrow. In contrast, multiple bright dots appeared on the DNA after resolution (71s incubation in the protein channel and 24s after braid resolution; fig. 1g).

I understand that the enzymes could constantly bind to and unbind from the DNA in the protein channel. But it does not explain a sudden increase in the number of proteins on the DNA for 24s after resolution. I wonder if it was due to a laser power setting change or scanning time change to enhance the detection after resolution. If it was the case, the authors underestimated the actual number proteins bound on the DNA before resolution.

TOP2 α resolution of DNA braids requires ATP hydrolysis

I agree with the authors that it is of interest how ATP hydrolysis step (binding, hydrolysis, and release) is coupled to the three-gates motion and DNA transfer. As described in the manuscript: "It is however unclear, whether ATP binding is sufficient for release of the transported T segment through the C-terminal gate. We sought to test whether DNA braid resolution requires ATP hydrolysis.", the authors intended to prove the T-segment release from C-gate would be hindered by ATPYS or AMPNP. However, as mentioned in my previous review, "no unbraiding" does not prove the thesis that ATP hydrolysis is required for releasing of transfer DNA from C gate. The lack of decatenation could mean three possible scenarios i) failure to capture transfer DNA (as ATP analogue binding may lock the gate before DNA capture), ii) failure to pass transfer DNA through the DNA gate, and iii) failure to release the transfer DNA from the C-gate after passage through the DNA gate. One way to confirm if and where T-segment is located within the protein could be done by either establishing that the DNA molecules are still linked by after mechanically reversing the crossing (transfer-DNA captured by enzyme; case of ii and iii) or by removing protein with SDS after decatenation test to see if the braid could be separated (iii).

The Diffusion measurements of TOP2 α with and without ATP showed some interesting results. However, the authors should take into account the residual flow effect in the cell that the authors dismissed as the flow caused only ~3 pN drift force significantly smaller than 60 pN.

TOP2 α resolves DNAs with multiple braids

Multiple braid resolution by different type IIA topoisomerases have been extensively investigated in previous studies at the single molecule level as listed in the references (ref 35, 36 and 38). Previous work elucidated the processivity of topo IIA on different braid geometry, chirality, and tension dependence. In this manuscript, the authors tested this under less rigorous condition where 4 braids were applied under high enzyme concentration without any crossing geometry information and a single poorly controlled force 5 ± 5 pN.

Fig.2e: Before full decatenation (middle image), three bright dots appeared in a row between two beads (bead 4 and 3) that almost looked like bound on DNA. After decatenation (image on the right), those dots still remained and one more dot appeared on the same horizontal line next to one of the previous dots. I am sure if those images actually corresponds to the decatenation activity detected by force change below.

Etoposide inhibits TOP2 α activity

The previous study by Le et al. (ref. 37) provided detailed mechanism of Etoposide dependent TOP2 α inhibition. At 50 μ M etoposide, reference 37 showed a minor reduction in the rate for human Top2 α compared to no etoposide as the etoposide dependent inhibition frequency was low (0.02-0.04 per turn). Interestingly, the current study showed 6/13 decatenation for single braid and 0/9 for 4 braids indicating that the inhibition frequency is >0.5 per turn. In my previous review, I asked the authors to explain what would cause such discrepancy. However, the authors response was somewhat confusing and even contradictory as below.

“As mentioned earlier, our four-braided substrate would require four TOP2a uncrossing cycles (not 2). However, we do not think that our results are inconsistent with those from Ref. 31 (Le et al. 2023). We have, therefore, stated that our results are compatible with what was reported in that study (Le et al. : page 6). Firstly, the magnetic tweezer experiments by Le et al. are carried out at 0.2 pN, not 5 pN. As shown in Figure 3, higher forces can affect TOP2a activity. Secondly, the data in Le et al. (Figure 6b, blue curve therein of their publication) does show the presence of a few small bursts of activity separated by very long pauses in the 300 s timescale, which are as long as our five-minute experimental window. Furthermore, in the Le et al. experiments, the DNA never returns to the initial length, indicating that etoposide does not allow TOP2a to fully uncross the DNA, in agreement with our results. Lastly, in the magnetic tweezer experiments by Le et al., the enzyme cannot be observed directly, and therefore, the reported activity could be the result of a multimer of TOP2a, rather than individual monomers (as suggested by Yogo et al.). “

I agree that higher forces affect Top2 α activity as shown here and in previous studies but we do not know if higher forces would also affect etoposide dependent Top2 α inhibition without further experiments. The authors seem to be speculating that force affects etoposide inhibition to a greater extent than Top2 α activity, which would be an interesting result, but would require substantially more force-dependent relaxation data to confirm.

The authors claimed that the data in Le et al. showed 300s pauses but figure 6b only showed an exemplary trace (Yeast topoll) and did not reflect an actual average pause duration.

Additionally, the enzyme concentration used in the study by Le et al. was 1 pM. Based on the study by Yogo et al, the data was highly likely taken with a single enzyme (Top2 α dimer). If multimer enzymes were present, I would expect more etoposide dependent Top2 α inhibition in terms of etoposide concentration compared to no etoposide condition. In line with this, 2 nM enzyme concentration (2000x more than one in the study by Le et al) used in the current study might explain higher inhibition frequency.

I do not find the information (“the DNA never returns to the initial length”) mentioned by the author in the study by Le et al.

TOP2 α DNA braid resolution requires controlled loading to the DNA junction

This is a potentially new result, but with the additional information provided in the rebuttal and revised manuscript I can identify two reasons why the conclusion from the results obtained in this study is less convincing.

First, as mentioned in my previous comment, it would be hard to draw a conclusion from 9-12 trials with 5 min measurement window for any Poisson process of which mean waiting time is significantly longer than 5 min. I agree with the authors that the enzyme binding at the crossings or crossing junctions should facilitate decatenation activity, particularly at high tension (5 pN) and a large crossing angle under which the diffusion of DNA crossing would be slow or limited. However, I am uncertain about how the authors drew the conclusion from the example of some small number of bound enzymes (~ 1 aM). As shown in the study by Yogo et al. (PLOS One, 2012), the waiting time to observe a first burst of activity was ~ 100 s even at 3.7 pM and no unlinking activity at 0.1 pM for 300 s cut-off time (interesting that the authors also chose the same time duration). This suggests that the reason for no decatenation for prebraided DNA in the current study could be just due to the extremely low probability to encounter a correct crossing geometry for decatenation even for Top2 α binding close to the junction.

TOP2 α resolves right-handed crosses more efficiently than left-handed

Chiral sensing by topoisomerase IIA is likely influenced by crossing angles rather than handedness itself as described in the previous studies (Stone et al. PNAS; Charvin et al ref 35.; Neuman et al. PNAS). The study by Yogo et al (2012 PLOS One) indicated that the crossing angle at 90° diminished the chiral preferences by TOP2 α . As such, it is difficult to conclude from the current measurement without well controlled crossing geometry of DNA braids. There is no description of the crossing angles imposed in the crossing in this manuscript, which makes the results difficult to interpret and difficult to compare to previous work in which the crossing angles were estimated.

TOP2 α DNA braid resolution is inhibited with force

As referred in the current manuscript (ref. 35 and 38), force-induced decatenation rate inhibition of topoisomerase IIA was previously investigated in detail. Based on the decatenation rate in terms of force (ref 38), the estimated waiting time for a single enzyme to decatenate would be ~ 32 s at 5 pN and ~ 1000 s at 10 pN indicating that the probability to decatenate at 30 pN within 5 min would be very low.

Cohesin inhibits TOP2 α DNA braid resolution

I agree with the assumption that TOP2 α decatenation activity would be hindered if other enzymes physically block the DNA crossing. As shown in the current study, not only Cohesin but also Condensin, which is known for enhancing TOP2 α decatenation activity in vivo (Dyson et al, EMBO 2021), could interfere with TOP2 α when they were bound to or near to crossing. However, the question is how prevalent the single braid with high crossing angle under high tension (5 pN) used in the study occur in vivo, perhaps, except for Anaphase. In particular, both SMCs could localize at the junction suggesting that the braid condition used in the current study might drive localization of SMC proteins at the crossing. Secondly, TOP2 α

could decatenate 3 out of 13 single braids (12/13 Cohesin junction binding) within 5 min suggesting that TOP2 α could overcome physical barriers and decatenate on the longer time-scale of cell-cycle steps.

In Summary, now that the authors have provided additional information and revised the manuscript in response the questions from the reviewers, my enthusiasm for the work has significantly declined. The majority of the results reproduce previous results, though with lower resolution, accuracy and experimental control, and the few possibly new results are open to interpretation given the experimental limitations. I cannot recommend publication in Nature Communications.

Reviewer #3

(Remarks to the Author)

I am happy with how authors addressed all my comments. One thing that I spotted in new data is I would recommend double checking the trap stiffness and state explicitly how signals in 2d,e were processed. Currently authors wrote that trap stiffness was 0.3 pN/nm. The first step in 2e is ~ 2 pN, which would make it ~ 6 nm and the noise less than 1 nm. Is this a bootstrapped data or how do authors get such high resolution? Also, even a single braid would probably be expected to be significantly longer than that given DNA persistence length on the order of 50 nm. This might affect interpretation. How much DNA length was resolved in a single 2 pN step and is it consistent with a fraction of braids? Also, what "relative force" means is unclear.

Version 2:

Reviewer comments:

Reviewer #1

(Remarks to the Author)

The authors have put in a lot of effort to address my concerns and raise the manuscript to a higher level. I am satisfied with the improvements they made, and can now recommend this manuscript for publication.

Reviewer #3

(Remarks to the Author)

New version of the manuscript provides additional interesting data, which I believe improved the overall story. From my perspective the manuscript is ready to be published.

I have read the communication with reviewer 2 and I think their comments are sufficiently addressed. There are obviously some questions about Topolla biology that remain to be studied in future, but overall, I believe there are sufficient important new advances in this work to warrant its publication in Nature Communications. Additionally, while some of the technical features of the commercial optical trap make is difficult to control all aspects of the experiment, overall, I am confident in the presented data and how it supports the conclusions of this work.

RESPONSE TO REVIEWERS' COMMENTS (NCOMMS-23-62652-T)

We would like to thank the reviewers for their insightful comments. We are glad that they found that “*the manuscript has the potential to provide valuable single-molecule insights into the function of TOP2a*” (Reviewer 1), “*presents interesting observations and potentially important insights on TOP2 α and cohesin interactions*” (Reviewer 2) and “*interestingly, authors show that in their conditions, where DNA is torsionally relaxed, TOP2 can resolve concatenations under loads*” (Reviewer 3). We have now addressed their suggestions point-by-point, as described below.

Reviewer #1:

Reviewer comment:

In this study, Erin E. Cutts and colleagues investigate the mechanism of TOP2a decatenation using braids of lambda DNA generated using quadruple-trap optical tweezers. They observe that TOP2a is able to untangle both single and multiple DNA braids, and that this is more efficient for right-handed braids than left-handed braids. Furthermore, they find that the untangling ability of TOP2a is inhibited by loading onto DNA in absence of a second DNA strand, as well as the presence of etoposide, non-hydrolysable ATP, and the presence of cohesin. Although the manuscript has the potential to provide valuable single-molecule insights into the function of TOP2a, there are some important concerns that should be addressed prior to its publication.

major comments

Fluorescence

I have some concerns about the quality and interpretation of the fluorescence micrographs. First of all, the signal of all used dyes appears to be very dim, making it difficult to see exactly what is happening in each image. Looking at the methods section, it seems that the authors used low fluorescence excitation laser powers. The authors should convincingly show that they have the single molecule resolution they claim to have (for example in line 275), using a study of photobleaching traces for all used fluorophores. If this analysis shows that single molecule resolution was not obtained, the experiments should be repeated under optimized conditions, or conclusions should be adapted accordingly. If/when single molecule resolution is obtained, it would be very insightful to convert the fluorescence intensity to the number of bound molecules, so conclusions could be made about the number of molecules responsible for certain events etc.

Our reply:

The reviewer is correct. We use low-excitation powers to reduce photobleaching. However, under our experimental conditions, we can readily resolve individual molecules. To address this issue, we have included a fluorescence intensity histogram of TOP2 α molecules at the DNA junction in the presence of ATP γ S (no decatenation) (Supplementary Figure 3c) and example photobleaching traces from TOP2 α molecules bound on dsDNA (Supplementary Figure 4d). The results confirm that we have achieved single-molecule resolution in our data. We have also clarified this and mentioned this data in the text (Pages 6, 7, 8).

Reviewer comment:

The labelling efficiency of TOP2a and especially cohesin is quite low. The authors should consider and discuss how this affects their data interpretation and conclusions, for example in line 275: 'it allows us to see the outcome of every TOP2A that binds'.

Our reply:

The reviewer is correct. It was a challenge to obtain higher labelling efficiencies. As suggested, we have further clarified this in the manuscript and discussed how this affects our interpretation and conclusions (Page 5).

Reviewer comment:

On several occasions the translocation of fluorescently labelled TOP2a is interpreted as diffusion. However, there isn't currently any analysis of the tracks of fluorescent particles to support this interpretation.

Our reply:

We thank the reviewer for this suggestion. We have now included a detailed diffusion analysis of TOP2 α in our manuscript (see Figure 2g, Supplementary Figure 4b-d, Page 7 and methods).

Reviewer comment:

Force

Both from cited sources and new data provided by the authors, it seems apparent that force greatly impacts TOP2a activity. However, the force data and its description presented in this work does not give sufficient information to truly appreciate the role of force in TOP2a-mediated braid resolution, and to rule it out as a confounding factor between experimental conditions. It is currently unclear in most cases from which bead the mentioned force is measured, and only an estimate of the applied force is given.

Our reply:

We apologize for this confusion. In our experiments, the force is measured on beads 1 and 4 (Figure 1). Both beads are calibrated by measuring their power spectra before DNA tethering, and the force is zeroed before measuring FD curves at a short bead-to-bead distance (where the force should be zero). We have now clarified this in the materials and methods (Page 16).

Reviewer comment:

The exact same force should be applied to the same bead for each experimental condition to obtain reliable results. Especially for the comparison to 'high force' of >30 pN the lack of precision is very problematic. The force should be well calibrated and kept constant during measurements.

Our reply:

Unfortunately, this is not practical for these experiments because the catenated DNA is prepared and validated manually. Furthermore, the C-trap measurements are much noisier under force-clamp mode than under constant-distance mode. Therefore, for

each experiment, we first calibrated the beads as described above, braided the DNA and then performed force-extension curves to confirm the presence of the braid. Finally, the two bead pairs were manually adjusted such that the force on the beads was around 3-7 pN and we checked that the force measurement on the second bead pair was 3-7 pN, and that the average across the bead pair was 5 ± 5 pN. However, within this range of forces, we did not observe any significant effects on TOP2 α activity. We have now clarified this in the materials and methods (Page 17).

Reviewer comment:

For force calibration of quadruple-trap optical tweezers, Brouwer et al., 2016, Optical Tweezers, Methods in Molecular Biology (https://doi.org/10.1007/978-1-4939-6421-5_10) might be a good starting point.

Our reply:

We thank the reviewer for this suggestion. We followed the calibration procedure recommended by the manufacturer, which is similar to the one described in the MMB publication suggested by the reviewer. However, a challenge we encountered for force calibration is that we must move the traps to the channel containing TOP2 α after calibration and braiding are done. Therefore, because of this movement across the microfluidics chip, the calibration is no longer accurate, but a close estimation of the actual force values (as explained in the MMB manuscript). We acknowledge that we should have been clearer in our initial submission, and we should have included this citation. We have now corrected this and included this and additional citations to other Q-trap papers in the updated manuscript (Pages 4 and 16).

Reviewer comment:

At minimum, a (supplementary) figure depicting the actual applied tension versus the resolution probability should be provided for each condition. Additionally, it would be better to give forces in figures as absolute instead of relative values.

Our reply:

As explained above, due to the limitations of the Q-trap system in obtaining accurate tension measurement at the braid when TOP2 α decatenation takes place (due to the necessary movement of the traps after calibration across different channels), we cannot provide exact force values for each braid. Instead, we state the force of resolution as an average of 5 ± 5 pN for braids tested when the DNAs are taut (in Figures 2a, 3b and 4d) and >30 pN for our experiments when DNAs are held at higher forces (Figure 3b; >30 pN).

Reviewer comment:

Timescales

For all of the investigated conditions, the timescale at which the braid resolution does (or does not) take place is quite relevant. Currently, there is no mention of these timescales. In my opinion, it

should at least be mentioned for how long each measurement was conducted and after how much time the resolution took place.

Our reply:

We agree with the reviewer, and we have now included the total resolution time in Supplementary Figure 3a. We find $\tau_{av} = 49.4 \pm 8$ s (N = 17) for a right-handed braid and $\tau_{av} = 48.3 \pm 12$ s (N = 8) for a left-handed one.

Reviewer comment:

Additionally, timestamps should be included instead of frame numbers for all fluorescence images.

Our reply:

We agree. Time stamps have been added in our data.

Reviewer comment:

Statistics

Throughout the manuscript, there are a few statements indicating the frequency of a particular event/finding, without mention of the number of events that support these claims. The N of measurements should be included in the following instances:

- Line 164: 'in many cases' - Line 196: 'in most cases'

- Line 221: 'TOP2a remained bound to the DNA after the resolution events' not added

- Line 236: 'kymographs show that...' No N given for number of kymographs or number of tracked proteins

- Line 320: 'kymographs show that...' No N given for number of kymographs or number of tracked proteins

Our reply:

We agree with the reviewer. We have tried our best to add N numbers throughout the manuscript. In addition, we now provide a table (supplementary table 1) with the N numbers and outcomes for all resolution experiments. The table is included in our Data Availability section.

Reviewer comment:

Some conclusions are supported only by slim statistics (N<10). Especially when statements are made that events never occur, the authors should take care not to overinterpret their data.

Our reply:

We acknowledge that experiments using optical tweezers are conducted one molecule at a time, which can sometimes limit the statistical power. However, we believe that our analysis has statistical validation. We have been careful not to overinterpret the results in our manuscript, as suggested by the reviewer.

Reviewer comment:

General

The authors should look more careful at the available literature on quadruple-trap optical tweezers experiments. This has been around for ± 15 years and quite some of the issues encountered here are solved and/or discussed before. They can thus provide answers and should be properly cited.

Our reply:

We apologize for the omission of some of the critical publications using quadruple-trap optical tweezers in the past. We have now included comprehensive literature citations on quadruple-trap optical tweezers experiments.

Reviewer comment:

minor comments

Line 35: here $>20\text{pN}$ is mentioned, whereas later in the text $>30\text{pN}$ is mentioned. Which one is correct?

Our reply:

We thank the reviewer for noticing this. The correct force should have been $>30\text{pN}$. We have corrected this mistake.

Reviewer comment:

Line 146: it would be useful here to mention that the activity of TOP2a was tested using decatenation assays.

Our reply:

We now mention this on page 5 in the revised manuscript.

Reviewer comment:

Line 150: ‘incubated the DNA with protein’: What was the incubation time?

Our reply:

We have now added “until resolution, or for 5 minutes”.

Reviewer comment:

Line 236: the reasoning behind the switch from Alexa555 to Cy3B should be given. Also, the labelling efficiency of TOP2a-CyB should be given in the methods section.

Our reply:

This was for practical reasons. We ran out of Alexa555 dye and tested an existing batch of Cy3B for the kymos. However, we feel this could cause some confusion for the readers, and hence have collected new diffusion data using the Alexa 555 labelled

material at 2 nM, the same protein concentration as used for the braid resolution assays. Diffusion analysis of both Cy3B and Alexa555 labelled TOP2 α suggest no difference, hence for clarity, we have only included the Alexa555 labelled kymos and analysis in the revision.

Reviewer comment:

Line 299: it is not immediately clear to me how this conclusion is in line with the data presented in this section.

Our reply:

We thank the reviewer for noticing this. This sentence has been deleted.

Reviewer comment:

Line 468: 'Pulling on DNA1 results in no force in DNA2'. Looking at figure 1d, the force in DNA2 in fact increases from 0 up to ~4 pN. How is this drift in force explained?

Our reply:

In these force-extension curves, the force in DNA2 seems to drift by a maximum of 3 pN, which is likely caused by a very small amount of residual flow in the cell. Nonetheless, this value is much lower than the 60 pN experienced by DNA1. We have now clarified this in Figure 1 legend and adjusted the graph scales to be comparable in Figure 1.

Reviewer comment:

Line 599: what was the reason behind the switch from potassium glutamate to sodium glutamate?

Our reply:

We had to switch to sodium glutamate to avoid potassium-induced precipitation with SDS. We have now clarified this in the text, on page 9 and in the Materials and Methods, on page 16.

Reviewer comment:

1.12 Line 629: when describing the fluorescence laser settings, both the wavelength and power (in mW) should be mentioned. Furthermore, details about the confocal imaging, such as pixel size and pixel dwell time should be given.

Our reply:

We agree with this reviewer. This information has been added to Materials and Methods on page 16 (Wavelengths are 488, 532 and 638 nm; Power is 0.2, 5 and 0.9 μ W, respectively; Pixel size is 100 nm² and pixel time is 0.1 ms).

Reviewer comment:

Figure 1b: the cartoon indicates overstretching, but no overstretching plateau is seen in the graph, as the graph only extends to 60 pN. Additionally, given the specifics of the DNA construct, overstretching in the cartoon should start from the ends, not the center.

Our reply:

We have modified the cartoon accordingly (New Figure 1b).

Reviewer comment:

Figure 1d: it is unclear whether the distance from bead 1-2 or bead 3-4 is plotted. It also could be misleading to the reader that the scaling on the y axis is quite different between both panels. Lastly, the symbols in the lower panel do not match the figure legend.

Our reply:

We have now added the distance between beads 1 and 2 to the Figure, rescaled the force axis and made the symbols to match. We have also rewritten the figure legend accordingly.

Reviewer comment:

Figure 2a: from this panel it is unclear that YOYO was present in the ATyS condition.

Our reply:

We have clarified this in the Figure.

Reviewer comment:

Figure 2d,e: it is not indicated what the dotted line in normalized fluorescence intensity plot represents. Additionally, it now appears that there is a time correlation between the fluorescence frames and the force plot, is this really the case?

Our reply:

In the figure legend, we have now clarified that the dotted line is the expected fluorescence intensity of a single fluorophore, based on a gaussian fit of the fluorescence intensities for the 4 braid experiment, we have added this data to supplementary Figure 3e. The scans are roughly correlated with the force vs time events, but time stamps have been added throughout for clarity.

Reviewer comment:

Figure 2e: in frame 2 there appears to be a DNA horizontally between the lower beads. Could it be that the braid was resolved during image acquisition?

Our reply:

The reviewer is correct with this assumption. We have included the time stamp which should clarify this. We also indicated this with a grey bar in the figure and highlighted this point in the figure legend.

Reviewer comment:

Figure 2e: if the braids are resolved in several steps instead of simultaneously, for each resolution only a miniscule amount of DNA would be released. How can this be unified with the significant force drops observed here? To help answer this it would again be useful to look at the absolute instead of relative forces.

Our reply:

Multi-step braid resolution is a relatively rare event (3/13 resolutions, Figure 2a). All three multi-step resolutions also exhibit two fluorescent complexes binding at the junction, as shown in Figure 2e. Therefore, we think that multi-step resolution events likely correspond to a small fraction of distributive resolutions. We are unsure what the first drop corresponds to, but we suspect it might be the first complex releasing the junction after the first resolution burst. We have now clarified this in the text (page 6). We have also added the force for the plots, as suggested.

Reviewer comment:

Figure 2f: it would be helpful for the interpretation to state the force at which the DNA was held for these kymographs.

Our reply:

We agree. We have now added it to the figure.

Reviewer comment:

Figure 4a: the DNA appears to be curved in this example. How were the kymographs generated from a curved molecule?

Our reply:

The reviewer is correct. We used a 10-pixel jointed line to generate the kymos in Fiji. We have emphasized this more explicitly in the Materials and Methods on page 17.

Reviewer comment:

For all figure panels containing fluorescence images, a scale bar based on the used pixel size should be included. Using the trapped beads as a scale is inaccurate and insufficient.

Our reply:

We agree. We have added scale bars to all the figures.

Reviewer #2:

Reviewer comment:

The manuscript by Cutts et al investigated human Top2 α decatenation activity and cohesin dependent regulation of Top2 α at the single molecule level in real time. The authors presented some interesting observations that could be of interest to the topoisomerase community. However, the measurements and the interpretation are somewhat flawed without time information. The authors modeled all the measurements as Bernoulli trials, treating each measurement as representing essentially a binary probability of success or failure. The decatenation activity by Top2 α , even with a single-braid, is difficult to characterize as a coin flipping event modelled as Bernoulli trial mentioned in the method without defining the time over which the trial is performed, which must be constant for each measurement.

As such, I would recommend the authors measure the waiting times from Top2 α localization at the DNA cross for individual decatenation events in order to characterize Top2 α activities under different conditions. Regardless of “not heterogeneous substrate”, the decatenation probability estimated based on the small number of events would be highly biased compared to the ensemble measurements using heterogenous substrates. A similar issue also caught my attention. The authors compared the decatenation activities among a few prebound enzymes before braiding and DNA braid with 1-2 nM of enzyme. Enzyme target finding process inherently takes time through a mixture of 3D and 1D diffusion (Halford et al; NAR 2004). Naturally, it takes longer to observe the activity with lower concentration of enzyme than higher concentration as previously tested at the single molecule level (Yogo et al; PLOS One, 2012) where no decatenation activity was observed at 0.1 pM for 300 s while <1s at 1 nM in the presence of 30 braids. Thus, reducing number of enzymes from 2 nM to a few should significantly increase the waiting time to observe decatenation activity, which makes unfair to conclude that preloaded enzymes become inactive.

The data shown in (Fig. 4) presented directly reveals that Cohesin can physically hinder Top2 α from accessing DNA crossing providing an insight on the regulatory role of cohesin in the separation of two sister chromatids by Top2 α . Taking advantage of the technique, it would be interesting to test the other possible Cohesin roles (Piskadlo et al; Int J Mol Sci 2017) – promoting Top2 α catenation activity by increasing proximity of two DNA. For example, the authors can test catenation activity by Top2 α using DNA configuration shown in Fig. 4f, panel 3.

Our reply:

The reviewer raises interesting points in their general assessment of our manuscript. We have addressed the majority of the points raised and provided specific actions and explanations to the individual concerns raised in our responses below.

With regards to the suggestion that Cohesin could promote the introduction of new catenations, we agree that this is an interesting possibility that merits further investigation. However, the main focus of this study has been TOP2 α decatenation and hence, we have only tested how factors such as cohesin affect this activity. We feel that in order to test the effect of cohesin on catenation by TOP2 α we would need to develop new assays that measure catenation and would need full characterisation of TOP2 α catenation activity in the absence of cohesin before being able to address how cohesin affected the activity. We feel this clearly goes beyond the scope and possibilities of the present study, however we will be investigating this in the future.

Reviewer comment:

The manuscript presents interesting observations and potentially important insights on Top2 α and cohesin interactions. However, there is a caveat in the measurements and the interpretation. Thus, without a major revision, I would not recommend it for publication in Nature Communications.

I would recommend the authors to include the following information for better understanding of the study:

1. Cut-off time before considering no decatenation, and more generally for every measurement. As it stands there is not a single-mention of the durations of individual measurements. Without this information, and unless the durations were identical, the results are extremely difficult to interpret. If the waiting time for each measurement was constant, then the authors have to clearly define the measurement approach, including how the durations were chosen, and why the more conventional and informative kinetic or dynamic measurements were not performed. Dependent on how long the authors waited before considering no decatenation, decatenation probability would change and hence, measuring the rates with repeated frequency are recommended. Furthermore, reducing kinetic measurements to probabilities makes comparisons to previous measurements, or interpretations of any implied kinetics, impossible. The authors should therefore provide a comparison between their observed probabilities and the probabilities that would be expected based on prior measurements of Top2 α activity, which will once again require well-defined and constant waiting time intervals.

Our reply:

We agree with the reviewer that time is an important parameter in these and all experiments. Our analysis was carried out using 5-minute experimental window, which is empirically determined to observe all resolution events for a single braid (Figure 2a). We have now clarified this in the methods section of the revised manuscript (Page 17) as well as the main text (Page 5). The reported resolution times are all from the moment the substrate enters the TOP2 α channel and we have added the distribution of these times for a single braid in supplementary Figure 3a. In most cases, the time between binding and resolution is too fast to be determined by confocal imaging (even with ~500ms frame time). Indeed, our results agree with those reported by Yogo *et al.* 2012 using optical tweezers. We acknowledge that we should have cited Yogo *et al.* in our paper. To address the reviewer's concerns, we have clarified these points in the main text (Page 5) and cited Yogo *et al.* (page 6).

Reviewer comment:

2. The average waiting times for decatenation events tested in the study.

Our reply:

We have now included all the waiting times (from entering the channel until resolution) in the manuscript (see Supplementary Figure 3a). However, the time between binding and resolution is too fast to be determined by confocal imaging (~500 ms frame time).

Reviewer comment:

3. The frequency of Top2 α localization at crossed junctions during decatenation measurements listed in Fig.2a. For example, If N=10, on how many occasions did the authors observe Top2 α localization at cross junction?

Our reply:

In most cases of non-productive resolution, there is always a TOP2 α molecule at the junction. In some cases, the resolution is so fast that we are not able to “detect” the binding preceding resolution. However, we agree with the reviewer that this information is relevant. We now included this data in Supplementary Table 1 in our Data availability section.

Reviewer comment:

Page 4; line 136-137: Could the authors clarify which characteristics of the braids generated by four beads would be similar to that of the catenates of two plasmids? Based on the simple geometric comparison, a crossing by the arrangement of four beads in this article should have larger crossing angles than those of two plasmids, which could affect efficacy of Top2 α decatenation activity.

Our reply:

We agree with the reviewer that the angles in our experiments could be larger than those in two plasmids. We have now highlighted this point in our manuscript (Page 5).

Reviewer comment:

Related to making a right-handed crossing, the description of the figure caption 1 c) “bead 3 and 4 are moved up in Z” doesn’t seem correct. It should be “lowering bead 3 and 4 in Z” relative to the plane of bead 1 and 2 in order to place DNA 2 beneath DNA1.

Our reply:

The reviewer is correct. We have now amended this typo in the revised manuscript.

Reviewer comment:

Fig. 1d DNA 2, triangle should be rectangular for DNA braid trace. And it would be clearer if different colors are used to distinguish Naked, braid, and DNA after TOP2A instead of different shades of red.

Our reply:

We agree with the reviewer. We have now changed this in the revised manuscript.

Reviewer comment:

I do not understand DNA 2 force extension curve. Shouldn’t the force and extension for DNA 2 remain constant when it was not braided or after decatenation by Top2 α when only DNA 1 was under a stretching force?

Our reply:

In these force-extension curves, the force in DNA2 seems to drift by a maximum of 3 pN, which is likely caused by a very small amount of residual flow in the cell. Nonetheless, this value is much lower than the 60 pN experienced by DNA1. We have now clarified this in the text and adjusted the graph scales to be comparable.

Reviewer comment:

Fig. 1e and g. Could the authors explain why more Top2 α fluorescent dots appeared on DNA after DNA unbraiding?

Our reply:

TOP2 α can still bind to dsDNA after braiding because it is still present in the channel. We have now clarified this in the main text (Page 5).

Reviewer comment:

Page 5; line 169- 183: The authors tested the idea that ATP hydrolysis is required for releasing of transfer DNA from C gate and concluded that the lack of braid resolution with ATPYS by Top2 α proved this idea. I think that the authors overinterpreted the result without obtaining direct evidence that the transfer strand is captured by the enzyme under these conditions. The lack of resolution could be that Top2 α bound at the crossing without decatenation. The only way to confirm capture of the T-segment in the C-terminal cavity is by either establishing that the DNA molecules are topologically linked by after reversing the crossing or by removing protein to see if the braid was resolved. T-segment capture in the C terminal gate in the presence of AMPPNP was directly observed by Martínez-García (NAR 2013). Additionally, there is some evidence showing that ATPYS is slowly hydrolysable, so AMPPNP would be a better ATP analogue for this assay.

Our reply:

The reviewer is correct that we don't observe the T-segment capture directly in these experiments. To address this, we have reworded the text accordingly to make it clear that this is a suggestion rather than a conclusion (Page 6). As suggested, we have performed additional experiments with AMPPNP, which yielded identical results as ATP γ S (see new Figure 2).

Reviewer comment:

Page 6; line 185-201: The authors investigated Top2 α resolution of multiple braid. Multiple braids resolutions by different type IIA topoisomerases have been extensively investigated in previous studies at the single molecule level by Yogo et al, Charvin et al (PNAS 2003) in addition to ref 32 that elucidated the processivity of topo IIA on different braid geometry, chirality, and tension dependence. In this manuscript, the authors replicated the previous works with less rigor.

Our reply:

We acknowledge that we should have included these citations in our initial submission. We have corrected this in the revised manuscript. We had to characterize the fundamental behaviour of the enzyme in our assay. Our work is in accordance with the published manuscripts. However, we go beyond the published work by imaging TOP2 α

during decatenation as well as describing the effect of cohesin in TOP2 α decatenation activity.

Reviewer comment:

4 braids is not enough to investigate processivity as it corresponds to two uncrossing cycles for topo IIA.

Also, given the relatively high enzyme concentration with one substrate in contrast to the previous works where the enzyme concentrations were as low as a few pM, one should expect multiple enzymes would potentially bind and resolve DNA braids. Perhaps, taking an advantage of fluorescently labeled Top2 α , it would be informative to measure an average processivity of a single Top2 α by applying high number of braids, which was difficult in other studies without fluorescent labeled enzymes. Additionally, it would be interesting to see where Top2 α prefers to bind along DNA braids. It has been suggested for topo IV to prefer at the end of braids while yeast topoll appeared to have no particular preference (Charvin et al; BJ 2005).

Our reply:

We respectfully disagree with the reviewer. Our four-braided substrate contains eight crosses (as shown in Figure 3a) and would require four TOP2 α uncrossing cycles (not 2). We already show the fluorescence data in Figures 2d and 2e, showing that four braids can be resolved by either one or two enzymes, which gives an idea of the processivity of TOP2 α . Regrettably, the assay cannot show more details about the kinetics of each uncrossing step. Unfortunately, we have not been able to apply a higher number of braids reliably with our instrumentation. While the binding position is an interesting question, we don't have the resolution to distinguish binding at the end or along the braid.

Reviewer comment:

Page 7; line 204-219: The authors investigated the inhibition of Top2 α by etoposide. However, considering the mechanism of etoposide binding, it is unclear why Top2 α could not resolve 4 braids that only takes two catalytic cycles.

As shown in the previous study (ref. 31), etoposide dependent blocking of catalytic cycle of topo II was transient and only slightly slowed down the catalytic rate with increasing etoposide concentration. At 50 μ M etoposide, reference 31 showed a minor reduction in the rate for human Top2 α compared to no etoposide suggesting etoposide dependent inhibition is transient, not indefinite. Could the authors explain what would cause such discrepancy?

Our reply:

As mentioned earlier, our four-braided substrate would require four TOP2 α uncrossing cycles (not 2). However, we do not think that our results are inconsistent with those from Ref. 31 (Le et al. 2023). We have, therefore, stated that our results are compatible with what was reported in that study (Le et al. : page 6). Firstly, the magnetic tweezer experiments by Le et al. are carried out at 0.2 pN, not 5 pN. As shown in Figure 3, higher forces can affect TOP2 α activity. Secondly, the data in Le et al. (Figure 6b, blue curve therein of their publication) does show the presence of a few small bursts of activity separated by very long pauses in the 300 s timescale, which are as long as our five-minute experimental window. Furthermore, in the Le et al. experiments, the DNA never

returns to the initial length, indicating that etoposide does not allow TOP2 α to fully uncross the DNA, in agreement with our results. Lastly, in the magnetic tweezer experiments by Le *et al.*, the enzyme cannot be observed directly, and therefore, the reported activity could be the result of a multimer of TOP2 α , rather than individual monomers (as suggested by Yogo *et al.*).

Reviewer comment:

Page 7; line 220-264: Based on numerous appearances of Top2 α fluorescent dots, it is possible that the reason for the failure of decatenation by bound proteins were not the same “active” enzymes that could decatenate shown in fig. 1e and 1g. Were those bound proteins able to diffuse along DNA? I think that the detection of diffusing Top2 α shown in fig. 2f were taken in 1-2 nM enzyme concentration. For rebraided (N=10) and preloaded (N=9) experiments, how often did the authors observe enzyme localization at the crossing and enzyme diffusion?

Our reply:

TOP2 α could be seen at the junction in 8 out of N=10 and 7 out of N=9 for rebraiding and preloading, respectively. We have added these numbers to the main text on page 7, and these observations are also included in Supplementary Table 1. We see diffusion in some cases, however, quantifying diffusion is not practical for the scan data, as the frame rate is too slow. We have quantified diffusion on a single piece of double stranded DNA (Figures 2f and g, and supplementary Figure 4b-d) and found that ATP binding can slow down diffusion, so we are not sure that the reviewers supposition that diffusive TOP2 α are active while static ones are inactive is correct.

Reviewer comment:

The authors suggested that the inability of preloaded Top2 α enzymes in the presence of ATP to decatenate was either due to wrong binding geometry or locked N-gate to capture of the second DNA. However, for wrong binding geometry argument, if Top2 α can diffuse, it could reorient itself in principle but it would take time to encounter crossing before diffusing off from DNA. For the ATP-dependent locked N-gate suggestion, I could argue that ATP hydrolysis will reset the blockage as Top2 α is able to undergo ATP hydrolysis with linear DNA. If ATP hydrolysis is hindered and N-gate is locked for some time, there would likely be differences in the enzyme diffusion rates with and without ATP as the enzyme would remain bound and less diffusive. In principle, the authors could measure them. I think it would be interesting to see.

Our reply:

To address this, we have quantified the diffusion of TOP2 α in the presence and absence of ATP, as suggested. Our data shows ATP does result in a small population of more slowly diffusing complexes, indicative of the conformational change, as the reviewer proposes. However, the enzyme mostly finds the braid by 3D diffusion, rather than by facilitated diffusion into a braid following binding away from it. We have now added the diffusion data in the revised manuscript (Figure 2f and g, and supplementary Figure 4b-d) and clarified this in the main text (Page 7).

Reviewer comment:

Page 8-9: The authors investigated if Cohesin could either block DNA separation after decatenation by Top2 α or prevent Top2 α decatenation activity all together. For measurements, the authors tried two different monovalent conditions. Could the authors explain why two different monovalent salt conditions were tested (K₂Glu and Na₂Glu)?

Our reply:

We had to switch conditions because potassium ions precipitate with SDS. We have now clarified this in Materials and Methods (Page 16) and the main text (Page 9 and 10).

Reviewer comment:

Cohesin can load on to a linear DNA easily but it could be inefficient to load onto and encircle a braid (Fig. 4e) as it has to capture both DNA within a pore. Alternative possibility would be dimerization of two Cohesins at the crossing. Could the authors distinguish these two possibilities?

Our reply:

We are actively exploring the question of Cohesin stoichiometry for a separate manuscript. Nonetheless, Richeld et al. NSMB 2023 have recently reported that a single Cohesin is sufficient to bridge two pieces of DNA.

Reviewer comment:

Minor point: “and” is missing in the sentence “direct interaction between Top2 α the BAF..”

Our reply:

We thank the reviewer for spotting this error. We have now made the correction.

Reviewer #3:

Reviewer comment:

In the manuscript by Cutts et al., the authors use optical tweezers to visualize how human TOP2 α enzyme enables passage of one DNA through another and in the case of one and four “braids”. Interestingly, authors show that in their conditions, where DNA is torsionally relaxed, TOP2 can resolve concatenations under loads of up to 30 pN. Finally, authors visualize how binding of cohesin to a DNA intersection prevents TOP2 from resolving it.

Major points

1. There seem to be several problems with an assay in which TOP2 resolves multiple braids:

a. TOP2 is expected to resolve multiple braids processively. However, authors report that in most cases multiple braids were resolved in a single step. Is this the result of a single TOP2 action? If yes, why doesn't a trace consist of multiple consecutive steps corresponding to resolutions of consecutive braids? Is the resolution of the optical trap sufficient to detect steps associated with decatenation of individual braids? If not, it should be straightforward to create more braids and visualize how multiple braids are resolved continuously.

Our reply:

The reviewer is correct, our data are also consistent with TOP2 α resolving braids processively. Previous studies suggest that processive braid resolution occurs in rapid bursts and pauses Le et al. 2023. The single-step (burst) force drop observed in our experiments likely corresponds to multiple processive resolution steps by individual TOP2 α enzymes, the resolution of the optical trap does not allow us to detect individual braids resolutions events in the single-step force drop dataset. The use of the word “step” in Figure 2 is probably causing confusion. To address this, we have changed it to “burst”, in line with other papers. We generate braids using an automated script, to ensure reproducibility. Unfortunately, creating more than 4 braids has been a challenge in our setup, hence limiting the number of braids that we can generate per experiment to 4.

Reviewer comment:

b. Does the same TOP2 molecule continue resolving consecutive braids after it binds the intersection, or do multiple TOP2 molecules exchange to resolve concatenation with four braids? Does the fluorescent signal come from a single or multiple TOP2 molecules?

Our reply:

We attempted to address this question in Figures 2d and 2e by plotting the fluorescence intensity of TOP2 α at the junction as it resolves the multiple braids. We have now added the corresponding distribution of fluorescence intensities across the 4 braid experiments in supplementary Figure 3e. We saw that all single-burst resolution events correspond to a single TOP2 α bound at the junction, whereas the multi-burst resolution events have at least two consecutive TOP2 α molecules binding at the junction. We, therefore, propose that all single-burst resolution events are mediated by a single TOP2 α molecule resolving consecutive braids. Unfortunately, even at the fastest 500 ms scanning time possible that we attempted did not allow us to determine

if there is enzyme exchange at the junction, as it might occur faster than this time scale. We have discussed these points in the main text (Page 6).

Reviewer comment:

c. Counterintuitively, several steps are indeed detected when brighter TOP2 signals are present (presumably due to multiple TOP2 molecules). How large are these steps and what is the distribution of their sizes? Are sizes consistent with a single or multiple braids being resolved? (Bottom figure 2e does not have numbers on y-axis and this information does not seem to be anywhere in the text). There is a contradiction in the data, which currently suggest that that multiple TOP2 resolve braids in multiple steps while at the same time one TOP2 resolves all braids in one step.

Our reply:

The reviewer is correct in that we do observe a small fraction of two-step resolution events (3/13, Figure 2). In these cases, we always observed two complexes at the junction. Since the data on single-step resolutions by individual TOP2 α molecules show that we cannot detect individual braids being resolved in our setup, we believe that these two-step resolutions are detected because the two TOP2 α complexes are not temporally coordinated, and the time separation allows us to detect the activities of the two TOP2 α molecules as two distinct steps. Unfortunately, the data does not allow us to reliably identify how many braids are resolved by each of the TOP2 α molecules.

We hope this clarifies the seemingly contradictory nature of the data, we have explained this in the manuscript (Page 6).

Reviewer comment:

2. It is an interesting observation that TOP2 can diffuse along the DNA, and it is surprising that diffusing molecules are much less efficient in resolving DNA braids. Authors conclusion that “TOP2 sits on DNA with geometry incompatible with the capture of the second DNA” is possible, but without any quantification or any other experiments to support this, a simpler explanation seems more likely. Decatenation efficiency depends on the on-rate with which TOP2 molecules bind DNA intersection. With 2 nM TOP2 in solution the on rate is likely much larger (given the vast number of molecules around available for binding) than the rate at which a single TOP2 diffusing on DNA can run into the intersection within the reasonable time of the experiment. This may also explain slightly higher efficiency of decatenation without ATP: although there is no quantification, fig 2f seems to suggest that there are more TOP2 molecules bound to DNA in the absence of ATP, and they diffuse more rapidly. This explains why single TOP2 molecules in the absence of ATP are more likely to run into the intersection and decatenate it, while there is more restricted diffusion in the presence of ATP, which makes these events extremely rare and probably unobservable within the timeframe of the authors’ experiment.

Our reply:

We agree with the reviewer on the possible interpretations. We indeed considered both explanations initially. We ruled out the possibility that decatenation efficiency solely depends on the on-rate for two reasons. First, because TOP2 α pre-loading in the absence or presence of ATP (Figure 2a) differentially affects the resolution efficiency, which would not be expected if diffusing to the junction was the rate-limiting step.

Second, in all our experiments, we give the molecules sufficient time to bind to the junction, which we checked using fluorescence, therefore ruling out that a diffusing enzyme did not have time to reach the junction. As suggested by the reviewer, we have now quantified the diffusion of TOP2 α in the absence and presence of ATP (Figure 2f and g, and supplementary Figure 4b-d). We have also clarified in the text that TOP2 α was observed at the junction in most cases where resolution did not occur (in “7 out of N=9 cases” - Page 8).

Reviewer comment:

3. More general comment related to the one above is that quantification of the binding/unbinding rates of TOP2 and its diffusion coefficients could help to distinguish between possible interpretations of data.

Our reply:

We agree with the reviewer, and we have now included this analysis in Figure 2f and g, and supplementary Figure 4b-d. As explained above, we were able to rule out diffusion as a rate-limiting step by confirming the presence of a fluorescent enzyme at the junction.

Reviewer comment:

4. Experiments in which authors show that cohesin blocks TOP2 activity support earlier studies proposing that cohesin protects centromeric regions from decatenations. The presented method of visualization does have the potential to reveal interesting molecular details, but they seem to be missing at this point.

a. Do authors suggest that the inhibition is purely mechanical? If yes, would any bulky protein prevent TOP2 decatenation, or does cohesin have any unique activity, which allows it to perform this specific function?

b. It is interesting that cohesin that can diffuse on one DNA and becomes trapped at the DNA intersection. Do all cohesins do that, or is that just that one example?

Our reply:

We agree that the mechanism by which cohesin inhibits TOP2 α resolution at the molecular level is an important question that is relevant to how sister chromatid cohesion (at centromeres) is maintained until the onset of anaphase.

Our data on TOP2 α decatenation activity (in the absence of cohesin) already indicates that the recognition of the crossed substrate is an important point that affects TOP2 α 's activity resolving braids, i.e. as highlighted by the decrease in decatenation efficiency when TOP2 α is preloaded on linear DNA before generating the braids (Fig. 2a). Therefore, we believe that a part of cohesin's inhibitory role is likely to be the selective binding to braided DNA to restrict TOP2 α from accessing these DNA substrates.

We have carried out additional experiments with condensin (a related SMC complex sharing many of cohesin activities, i.e. loop extrusion) which are consistent with this possibility. We show that cohesin selectively binds to braids and remains stably

attached to them (Fig. 4a and c), while condensin binds but does not show stable binding over time (Fig. 4b). Consequently, when we measured resolution by TOP2 α in the presence of these two SMC complexes and we observed that cohesin is substantially more inhibitory than condensin (Fig 4d).

With regards to cohesin diffusing to the DNA intersection, although in the original submission we provided an example of cohesin diffusing and binding to the junction (Figure 4, example 1), we did not intend to infer that cohesin molecules have to associate with junctions by diffusing on DNA until they meet these structures. In fact, the vast majority of cohesin molecules observed in our experiments bound quickly (and potentially directly) to the junction. We have a second example of cohesin at the junction when we start imaging (example 2). The technical challenge is that we must position the beads in an unusual geometry to generate kymographs with sufficient time resolution (Figure 4a).

Reviewer comment:

If cohesin has ability to track and block DNA intersections wherever they go, that would be an interesting mechanism for TOP2 inactivation, but I am not sure if authors' results are sufficient to claim this or even if that's what they aimed to do. Some quantitative data would be needed to support or rule out this model.

Our reply:

The issue raised here by the reviewer comes back to the original role of DNA catenations in providing sister chromatid cohesion. Cohesin was identified in genetic screens as a factor that caused reduced sister chromatid cohesion when inactivated. Over the last decade, most studies trying to clarify this cohesion role assumed that cohesin might physically hold sister DNAs and considered that intertwines between sister chromatids did not contribute to sister chromatid cohesion because robust cell-cycle regulation (activation/inhibition) of TOP2 α activity was not found. However, it is possible that cohesin's role in cohesion might be two-fold, one to physically associate and hold DNA intersections between sister chromatids, and two, inhibiting TOP2 α resolution at these sites. This idea is supported by studies demonstrating that centromeric catenations are protected by cohesin (as highlighted by this reviewer earlier) and a more recent study (Cameron *et al.* 2024 *Science*) showing that cohesin is pushed by the replisome to sites of DNA replication termination, which are known to be the sites where DNA intertwining occurs (Sundin and Varshavsky, 1980 *Cell*).

We feel that it is unlikely that cohesin acts as a general mechanism for TOP2 α inactivation, but rather it operates by protecting catenation between sister chromatids at discrete genomic sites. We feel that these are complex questions that require significant further analysis and are therefore beyond the scope of this manuscript.

Reviewer comment:

c. What is the topological state of cohesin at the intersection in these experiments? Does its ability to block decatenation by TOP2 depend on whether it is topologically or not topologically bound?

Our reply:

The issue of topological entrapment is a highly debated point in the cohesin field. Some studies suggest that resistance to high salt indicates topological binding, however this view has been challenged by the Nasmyth laboratory, which argues that topological loading can only be assayed by artificially crosslinking all SMC-kleisin gates (something very challenging to test in our tweezers experiments). In our experimental conditions, cohesin binding to both linear DNA and DNA junctions (where one DNA is overlaid on top of the other; note this is different to a DNA braid) resist high-salt. However, given the state of the field it is uncertain whether this is equivalent to topological binding. This salt-resistance data is part of a manuscript in preparation focused on cohesin.

Reviewer comment:

d. Does the cohesin inhibition depend on tension?

Our reply:

Because TOP2 α 's intrinsic activity is strongly impaired by force (Figure 3b), it would be challenging to untangle an additional force effect on cohesin's function. Increasing the force would only decrease the resolution probability further from an already low probability, Figure 4b.

Reviewer comment:

Minor

1. Fig. 1. What are the on and off rates of TOP2 on single DNA versus DNA intersections?

Our reply:

Unfortunately, we cannot calculate the off rates for TOP2 α because we never see it dissociate in the five-minute experimental window.

Reviewer comment:

2. Fig. 1f. doesn't have numbers on y-axis. What are the steps?

Our reply:

We have restored the y-axis numbers in Fig. 1f.

Reviewer comment:

3. Representation on Fig2a is misleading (Same for Fig. 4b). First bar is 8/8, which should be one. Why does it correspond to 0.9 in the left axis? Same for 0/9, which should be zero, but it is ~0.1 on the axis.

Our reply:

We performed an error estimate based on Bayesian inference analysis, treating each experiment as a Bernoulli trial using the beta distribution as described by Kinz-Thompson *et al.* 2021 (see Methods). But this was probably not clear enough in the manuscript. We have now clarified this further in the revised manuscript (Page 5) and in the corresponding Figure 2, 3 and 4 captions.

Reviewer comment:

4. It is unclear how error bars were calculated. SD for a binomial (or beta) distribution with the mean equal to zero is also zero. What do then error bars on 0/14, 0/9, 0/10 and 0/9 represent? What is the point of using error bars anyway if exact numbers are given? It may be more useful to have a measure of statistical significance of the difference between the experiments.

Our reply:

We performed an error estimate based on Bayesian inference analysis, treating each experiment as a Bernoulli trial using the beta distribution as described by Kinz-Thompson *et al.* 2021 (see Methods). We provided the N for experiments, in line with Nature communications editorial policy. We have now clarified this further in the revised manuscript (Page 5) and in the corresponding Figure 2, 3 and 4 captions.

Reviewer comment:

**5. Figure 2d and 2e middle graphs. What should this fluorescence intensity be compared to? What is the expected signal from a single fluorophore?
I guess I need a bleaching trace.**

Our reply:

To illustrate the expected signal from a single fluorophore, we have added a distribution of fluorescence intensities in supplementary materials (see Supplementary Figure 3e).

Reviewer comment:

6. Figure 2d and 2e bottom graphs. Units on y-axes are unclear. How big is one step? What is the distribution?

Our reply:

We have restored the y-axis numbers. Step sizes vary significantly depending on the initial geometry of the beads, which can vary significantly from one experiment to the next.

Reviewer comment:

7. Fig. 2f, do TOP2 on and off DNA binding rates and diffusion rates depend on its ATP state?

Our reply:

We have added this analysis (Figure 2f and g, and supplementary Figure 4b-d).

Reviewer comment:

8. Assuming K_d of etoposide is in the 1 μM range, concentration of the drug of 50 μM and TOP2 is 2nM, the ratio between the drug bound and unbound forms of TOP2 should be ~ 50 . Authors report that efficiency of DNA cleavage for one braid drops ~ 2 times when 98% of TOP2 molecules are presumably inhibited. Is this a contradiction?

Our reply:

Etoposide only binds TOP2 α after the first DNA cleavage event, but not TOP2 α in solution (Figure 2c). Hence, if the rate of strand passage and religation is faster than etoposide binding, a single-braid resolution can be allowed. However, multiple-braid resolution becomes exponentially inhibited. This is why we can see 50% resolution when TOP2 α activity is tested on a single braid, but essentially no resolution when 4 braids are used. This may not have been clear enough in the initial submission of our manuscript. To address this point, we have clarified this in the main text (Page 7).

Reviewer comment:

9. What counts as a number of trials in quantifying decatenation events (Fig. 2a)? Presumably these are the events in which TOP2 was visibly bound at the intersection. How long does TOP2 needs to stay at the intersection to be counted as a single trial? How far can it bind? What was the off rate for TOP2 in unsuccessful events?

Our reply:

Reviewer 2 also raised a similar point. We acknowledge that the original submission did describe this with insufficient clarity. We have now expanded the definition of a successful trial in the Methods section (Page 17) and in the main text (page 5).

Regarding the off rate for TOP2 α , we never clearly observe TOP2 α to dissociate from the junction once it binds. It simply fails to resolve it.

Reviewers' comments:

Reviewer #1 (Remarks to the Author):

1. General remarks:

Although the authors have managed to improve the manuscript in some important ways, in my opinion there are still remaining structural flaws (see comments below). More generally, the manuscript feels kind of rushed and should spend more attention to details. Therefore, in my opinion, the manuscript is not yet ready for publication in Nature Communications.

We thank the reviewer for the detailed feedback and for acknowledging the improvements we have made to the manuscript. We appreciate the constructive comments and have carefully considered each point to further improve our work. Below, we address the concerns, we have also revised the manuscript to improve the overall attention to detail.

2. Regarding the answer on our previous concerns on Fluorescence:

It is difficult to compare intensities in figure Supplementary Figure 3c and Supplementary Figure 4d, as intensities in Supplementary Figure 3c were normalized. In the current state, I don't see added value in Supplementary Figure 3c. It is unclear if the lowest normalized intensity in Supplementary Figure 3c corresponds to single molecules, for example.

We understand the confusion regarding the comparison of intensities between existing figures Supplementary Figure 3c and Supplementary Figure 4d. Our intention was not to have these datasets directly compared, and we apologize for any misunderstanding caused by quoting these figures together in our response to the reviewer. Our intention was to illustrate that we are capable of detecting single TOP2 molecules. If single molecules were being detected we would expect two distinct fluorescence intensities, for single and double labelled dimers, and given the labelling efficiency, we would expect the single labelled population to be approximately 4 times larger. We would also expect to see bleaching steps in the intensity of the kymographs. We felt these two figures, despite the different data units, illustrated this, however, we have now provide the measure of fluorescence intensity in the same unit (photons) for both figures, and they are roughly comparable. We would encourage caution when comparing of data from a scan and kymo. Intensity fitting of a scan uses a 2D Gaussian, of a stationary TOP2, resulting in a well described shape, that is amplitude and sigma, while kymos are the result of a single slice through the point spread function of the fluorophore on a moving TOP2, hence may not perfectly align with the highest intensity region and be more prone to noise.

3. Line 182-184: “Although we observed TOP2 α localisation at the DNA junction, no resolution events were observed when ATP γ S was present (N=14, Fig. 2a, Sup Fig. 3b and c).” Supplementary Figure 3c does give any information to support this claim. Figure S4d is not mentioned in the main text where it is relevant in this context.

The purpose of Sup. Fig 3c (new Fig 3b and Sup 4a) is to provide an example of localisation at the junction. We have now clarified that and provided N number directly observed, we have added the following sentence: “We used 1mM ATP γ S, a non-hydrolysable ATP analogue, in our assays instead of ATP and observed no resolution events (N=0/14, Fig. 3a), despite observing TOP2 α -AF555 localised at the DNA braid junction in all cases (N=14/14, Fig. 3b, Supplementary Fig. 4a).”

A reference to Fig S4d has been included in the photobleaching discussion: “Kymograph analysis shows that TOP2 α -AF555 molecules not only bind but also freely diffuse along dsDNA and this was observed both in the presence and absence of ATP (Fig. 3d and e) with fluorescence intensity and photo bleaching steps consistent with TOP2 α dimers (Supplementary Fig. 4d).”

4. The diffusion analysis (especially the figures) is not well explained appears incorrect. Some examples: In Supplementary Figure 4c there are a lot of data points within 1 second, but this does not seem to make sense given the line scan time. In Figure 2g one of the calculated values for D is negative. See King et al. 2018 Nucleic Acids Research for an example of diffusion analysis on optical tweezer data. Diffusion analysis should also be performed for cohesin, as the authors mention this molecule also diffuses along the DNA (Fig. 4a).

We appreciate the reviewer’s feedback on the diffusion analysis and have added extra details to clarify this. Our approach is the same as in King et al. 2018 in Nucleic Acids Research, where point correspond to frame rate. In King et al the frame rate is 0.5s and in our Kymos the frame rates (line time) are “60.5 ms for TOP2 α kymos and 57.9 ms for cohesin kymos”.

The diffusion data in Figure 2g (new Figure 3f and g) is plotted and Gaussian fitting is performed on a log scale, hence slowly diffusing molecules, ie with MSD less than 1 kbp 2 s $^{-1}$ have negative Log(D). The diffusion coefficient is now quoted in text “TOP2 α in the absence of ATP had a single diffusion coefficient of 1.9 ± 0.2 kbp 2 s $^{-1}$ (Fig. 3f and Supplementary Fig. 3e and f, N=29), while in the presence of 1 mM ATP there were two populations, with diffusion coefficients of 0.4 ± 0.1 kbp 2 s $^{-1}$ and 3 ± 0.2 kbp 2 s $^{-1}$ (Fig. 3g, N=33).”

As suggested, we have now included diffusion data for cohesin in the manuscript. This additional analysis can be found in the updated Figure 5b and Sup Figure 5a and b.

5. Regarding the answer on our previous concerns on Force:

5 ± 5 pN is a very strange force range to give.

The authors still do not show any improvement for the high force (>30 pN) case, where their argumentation about noise during force clamp will not hold. The main problem here is that only a lower threshold is given. With this definition, experiments in this condition could have been performed over a force range of tens of piconewtons, making the data impossible to interpret. Once again, the force should be as similar as possible between measurements of the same experimental group. In a force clamp mode where the force is set by the user, this really should not be a problem.

Our earlier reviewer comment:

At minimum, a (supplementary) figure depicting the actual applied tension versus the resolution probability should be provided for each condition. Additionally, it would be better to give forces in figures as absolute instead of relative values.

your reply:

As explained above, due to the limitations of the Q-trap system in obtaining accurate tension measurement at the braid when TOP2a decatenation takes place (due to the necessary movement of the traps after calibration across different channels), we cannot provide exact force values for each braid. Instead, we state the force of resolution as an average of 5 ± 5 pN for braids tested when the DNAs are taut (in Figures 2a, 3b and 4d) and >30pN for our experiments when DNAs are held at higher forces (Figure 3b; >30pN).

Our answer to this:

See comment above. No improvement is given for the high force case. This remains an important point to improve.

We thank the reviewer for the insightful comments. We have now included data determining mean resolution efficiency at fixed forces of 5, 15, 25, 30 and 45 pN, provided in Figure 2a. The force is applied in the buffer channel, hence is well zeroed and the geometry is fixed while moving into the protein channel. We have used the force from the first 10 low frequency data points to calculate the average force measure on bead 1 and 4 and the estimated crossing angle in Figure 2e. This results in mean forces ± standard errors of 4.8 ± 0.5 , 14.6 ± 0.5 , 24.8 ± 0.2 , 30.7 ± 0.8 and 44.8 ± 0.9 pN. In this data we still reduction in resolution at forces higher than 30pN, however see no difference in mean resolution between 5 and 15 pN. This is consistent with our previous result and we can also conclude that existing results where the force had a larger error, ie 5 ± 5 pN did not influence the results.

We thank the reviewer for this suggestion, and think the new result adds much to the paper. We have added further discussion as to why we think TOP2A fails at forces higher than 30pN and compared the results with existing studies where resolution rates have been measured.

6. Our earlier reviewer comment:

1.12 Line 629: when describing the fluorescence laser settings, both the wavelength and power (in mW) should be mentioned. Furthermore, details about the confocal imaging, such as pixel size and pixel dwell time should be given.

Our reply:

We agree with this reviewer. This information has been added to Materials and Methods on page 16 (Wavelengths are 488, 532 and 638 nm; Power is 0.2, 5 and 0.9 μ W, respectively; Pixel size is 100 nm² and pixel time is 0.1 ms).

Our answer to this:

Instead of μ W the authors probably mean mW.

Pixel size is likely 1000 nm², corresponding to a pixel of 100 nm by 100 nm.

We have made these changes:

“Images were acquired with a pixel size of 100 nm x 100 nm, a pixel time of 0.1 ms, with 1% green laser (λ_{ex} = 532 nm at 5 mW), and 2% red and blue laser (λ_{ex} = 638 and 488 nm at 0.9 and 0.2 mW, respectively).”

Reviewer #2 (Remarks to the Author):

1. As additional information has emerged in the rebuttal provided by the authors response to the reviewer’s critiques, the overall strength of the manuscript has decreased and I am less certain about the results, the conclusions, and the suitability of the manuscript for publication in Nature Communications. For example, the stated protein labelling efficiency dropped from 70% in the original submission to 30% in the revision. Furthermore, the relatively slow scan rate (0.1ms/100nm²pixel) was only reported in the revised manuscript along with the unusually large uncertainty in the force (5 ± 5 pN). Overall this new information and the discrepancies between the initial submission and the revised submission have diminished my confidence in the results and conclusions and therefore my enthusiasm for the work. This critical information should have been included in the initial submission, which indicates a lack of adequate rigor.

We appreciate the reviewer's evaluation of our manuscript. However, the assertion that new information has emerged that reduces the validity of our work, and the suggestion that it was previously concealed is incorrect. All information provided in the revised manuscript was either an expansion upon the initial data or additional details requested during the review process, and most importantly it does not invalidate any of our previous results. Suggesting otherwise implies a misunderstanding or an unfair interpretation of our work and intentions. We have

been transparent throughout this process and committed to providing all relevant data and clarifications requested by the reviewers.

We would like to clarify and correct some misunderstandings, and address the assertion of a lack of rigor in our work;

1. Contrary to the reviewer's comment on labelling efficiency, we did state in our original manuscript that the initial labelling efficiency was 30% for the preparation of protein labelled with AF555. However, we purified a second batch of protein, labelled with Cy3B in which the labelled efficiency was improved to 70%, as described in our original submission: "*Labelling efficiency were ~30%, and improved to ~70% for TOP2 α by enriching using the Strep tag after labelling prior to gel filtration.*" The higher labelled material was used for diffusion studies, where dark proteins could influence the results. For our revised manuscript, we acknowledge that the use of two different dyes was confusing hence recollected with the first prep of 30% AF555 protein, hence removed mention of the Cy3B labelled material and simply wrote: "*Labelling efficiency of TOP2 α Alexa555 and cohesin STAG1-ATTO647N were ~30%.*" Moreover, in this current revision, we required additional protein, hence did a new protein prep which was labelled with AF555, adding the additional enrichment step, to obtain a labelling efficiency of 70%. In all cases, the reported protocol represented the reported data, and the protocol only changed to reflect the change in the data presented. Control experiments carried out using the protein batches with different labelled efficiencies (in each of the three reviews) yielded identical results with regards to resolution statistics, strengthening our results, further demonstrating, that the labelling efficiency does not affect the conclusions of our study and should not be considered a reason to discredit our analysis.
2. The scan rate reported is the standard when using confocal microscopy in our optical tweezer instrument (C-Trap LUMICKS). We included this information in the revised manuscript in response to one of the reviewer's comments on imaging. Importantly, the scan rate does not impact the overall conclusions of our study. Even if a faster scan rate were possible, it would not improve the data, as our analysis does not rely on any assumptions affected by scan rate. Therefore, as in the case of labelling efficiency, scan rates are not a reason to discredit the conclusions of our study.
3. As explained in our response to reviewer 1 above, the unconventional force range of 5 ± 5 pN presented in our revised manuscript was given to provide a broad error margin rather than reflecting uncertainty on the forces used. As mentioned previously, our experiments involve calibrating the force in one channel before transferring DNA samples to another channel. This transfer, along with the accompanying flow, causes minor variations in the applied force. We recognize the importance of investigating the effects of force, and have collected additional data at carefully controlled forces of 4.8 ± 0.5 , 14.6 ± 0.5 , 24.8 ± 0.2 , 30.7 ± 0.8 and 44.8 ± 0.9 pN. Please see response to reviewer 1.5.

Finally, we respectfully disagree with the implication that the responses in our revised manuscript related to the points raised by the reviewer here (i.e. labelling efficiency, forces and scan rates) demonstrate that our work lacks rigor is an

unfounded evaluation by this reviewer. From the initial submission through the revisions, we have strived to present our findings with clarity and integrity. The revisions were made to enhance the manuscript's transparency and comprehensiveness, and did not reveal any obscure aspect of our work presented in the original submission as implied by the reviewer.

2. The revisions and rebuttal also reveal that the vast majority of the results are simply recapitulations of previous measurements, albeit with a new technique that generally appears to produce inferior data. It would be reasonable to include one figure describing “proof-of-principle” measurements reproducing previous studies for the purpose of showing the robustness of the enzyme and experimental approach. However, most of the TOP2 α decatenation results do not provide any new information and rather reconfirm previously established results “qualitatively” or produced different outcomes compared to previous studies for which the authors did not provide a reasonable explanation for the discrepancies.

While we appreciate feedback we respectfully disagree with the assertion that the majority of our results merely recapitulate previous measurements and that our new technique produces inferior data. This evaluation overlooks several significant and novel findings that are critical and will contribute significantly to our understanding of Top2 α decatenation.

First, we demonstrate that Top2 pre-bound to DNA is inefficient in resolving catenations, highlighting the crucial importance of the substrate recognition step. This insight into the enzyme's mechanism has not been previously reported as previous studies have not visualised labelled protein and provides a deeper understanding of how Top2 interacts with DNA.

Second, our study identifies a force threshold above which Top2 cannot decatenate DNA. This is clearly illustrated in our new data and represents a significant advancement in the field. This finding is particularly important as it has major implications for chromosome decatenation during anaphase when such forces are typically encountered by segregating chromosomes.

Thirdly, our study demonstrates that cohesin associated with DNA braids inhibits Top2 resolutions, suggesting cohesin has a role in protecting catenations from Top2, which may contribute to cohesion.

The only previously established point we confirm is the chirality preference of Top2, which we also observe. This serves to validate our experimental approach rather than to merely replicate existing knowledge.

The suggestion that our work lacks novelty seems aimed at discrediting our contributions without a thorough and fair assessment of the data presented. We

believe our findings provide substantial new insights into the mechanisms of Top2, and we have ensured that our manuscript clearly communicates these advancements.

3. Moreover, the reliance on time-limited trials to measure enzyme activity is a step back from the real-time kinetic measurements that are now commonplace in current single-molecule measurements of topoisomerase activity (e.g. references ref. 35, 37 and 38).

Our study indeed differs from previous investigations, which primarily focused on the enzyme's rate of activity and use magnetic tweezer based technology (where protein imaging is not possible). Our approach instead examines the probability of resolution. However, we acknowledge the importance of contextualizing our findings with existing literature. To this end, we converted our data at 5 pN to measure the rate of enzyme activity and provided further discussion in the manuscript: *“an estimate of 106–200 s per resolution can be obtained assuming a single-exponential process, using the probability of resolution across different wild-type conditions at 5 pN and the 5-minute sampling window”*.

This conversion reveals that the rate we observed is very similar to those reported in the studies referenced (references 35, 37, and 38), which validates our methodology and findings, rather than making them inferior, as wrongly suggested by the reviewer.

Moreover, our work extends beyond these traditional measurements by exploring the impact of high forces on topoisomerase activity, something that was not experimentally tested in the references quoted by the reviewer. Specifically, we demonstrate that when forces are between 20 and 30 pN the force significantly influences enzyme activity. This observation it is likely due to the force exerted on the substrate, and adds a novel dimension to our understanding of topoisomerase function that has not been previously reported.

Given that our work introduces new and significant information about the behaviour of topoisomerase under high force conditions, we believe it is incorrect to consider our approach a step back. Instead, it complements and enhances existing knowledge, offering a more comprehensive view of the enzyme's behaviour under varying conditions and using different technical approaches.

4. Thus the majority of the TOP2 α decatenation results reproduce prior measurements whereas the potentially new results (cohesion-dependent TOP2 α inhibition) are not performed with sufficient rigor or precision to justify the novel conclusions.

As mentioned in our previous point, our work on TOP2 α decatenation provides new insights that are crucial and not addressed in previous studies, particularly regarding

substrate recognition and high force inhibition. Therefore the findings add a valuable dimension to the existing body of knowledge.

Regarding the comments on the rigor and precision of our work on cohesin-dependent TOP2 α inhibition, we believe this statement is unfounded. We would appreciate specific feedback on why our work lacks rigor or precision so that we can address the issues. Our experiments were conducted with meticulous attention to detail, following standard protocols and controls to ensure the reliability and validity of our results.

5. The results relating to cohesin dependent TOP2 α regulation could be of interest, but the high force (5 pN or higher), large force uncertainty (100%), and ambiguous, poorly controlled, crossing geometry, which has been shown to influence TOP2 α activity, with relatively short measurement duration trials (5 min) render the main conclusion circumstantial or ambiguous.

Firstly, we believe that our work on cohesin regulation of TOP2 α activity is novel and of significant importance. Concerning the force used in our cohesin experiments, while the 0-10 pN force might seem high to the reviewer, it is important to note that TOP2 α resolution efficiency remains unaffected within this force range (Fig. 2). Our approach to testing TOP2 α activity within a 5-minute window is based on controlled experiments that demonstrate the probability of resolution in our assay is not directly related to the duration of the measurement. This can be clearly observed in our new analysis of resolution at different forces (Fig. 2).

Furthermore, our experiments using cohesin were repeated in two different buffer conditions, using either potassium and sodium glutamate as salt, yielding comparable results. Therefore, we believe it is inaccurate to characterize our main conclusions as circumstantial or ambiguous.

6. Overall, I found that the revision would require significant improvements in data quality, control of experimental variables (force, crossing geometry), and statistics, not to mention kinetic measurements of enzyme activity. Additionally, the authors' responses to my major concerns were often unsatisfactory and even contradictory. Regrettably I cannot recommend the manuscript to be published in Nature Communications.

We regret that our previous responses did not fully address the concerns raised. We have now made significant improvements to the manuscript by thoroughly addressing all the points mentioned, providing additional controls for key experimental variables, such as force and crossing geometry.

Given these comprehensive revisions, we believe we have adequately addressed all major concerns. We respectfully ask the reviewer to consider these improvements in their assessment. We trust that the reviewer will be fair and guided by facts in re-evaluating our manuscript for publication.

Specific points

6. The labeling efficiency information confusion:

In the previous version, “Labelling efficiency of TOP2 α Alexa555 and cohesin STAG1-ATTO647N were ~30%, and improved to ~70% for TOP2 α by enriching using the Strep tag after labelling prior to gel filtration.”

In the current revision, “Labelling efficiency of TOP2 α Alexa555 and cohesin STAG1-ATTO647N were ~30%.”

70% to 30% labeling efficiency would change the fraction of TOP2 α dimers containing at least one labeled monomer from 90% to 50%, thus significantly reducing the ability to directly observe TOP2 α dimers bound to the DNA, and increasing the likelihood of “dark” proteins acting on the DNA.

Due to the low labeling efficiency and low temporal and spatial resolution, the detected TOP2 α at the crossing might not be the one that is active. This also raises questions related to the coupling between TOP2 α localization at the crossing and decatenation activity.

We explained in a previous response the context of the reduced labelling efficiency in our resubmission.

We understand that a reduction in labelling efficiency might change the fraction of TOP2 α dimers containing at least one labelled monomer. Nonetheless, this does not significantly impact the conclusions of our experiments. As stated in the manuscript, we observe a clear TOP2 α signal near DNA crossings, providing direct evidence that the enzyme is detected at the relevant sites of activity. Additionally, the majority of our experiments focus on measuring resolution probability rather than quantifying the absolute number of active TOP2 α molecules at the cross site. Consequently, even if some TOP2 α molecules are not labelled, this does not critically affect the accuracy of our resolution probability measurements. The quantitative analysis of the number of proteins, which was introduced at the request of Reviewer 1 during resubmission, is limited to Fig. 3c and 4d-f and i, and Sup. Fig 4d, h and i.

While we acknowledge that single-molecule experiments inherently carry the risk of missing individual proteins, this is a common challenge in the field, and we have addressed it by accumulating sufficient data and employing statistical methods to minimize errors. Our approach adheres to best practices in single-molecule studies, and we are confident in the statistical robustness of our data.

7. In line with this, I previously asked why more enzymes seemed to appear after braiding resolution (fig. 1e and g). However, the authors reply (below) does not fully explain:

“TOP2 α can still bind to dsDNA after braiding because it is still present in the channel. We have now clarified this in the main text (Page 5).”

Before resolution (47s incubation in the protein channel; fig. 1e), there was one small bright dot indicated by the arrow. In contrast, multiple bright dots appeared on the DNA after resolution (71s incubation in the protein channel and 24s after braid resolution; fig. 1g).

I understand that the enzymes could constantly bind to and unbind from the DNA in the protein channel. But it does not explain a sudden increase in the number of proteins on the DNA for 24s after resolution. I wonder if it was due to a laser power setting change or scanning time change to enhance the detection after resolution. If it was the case, the authors underestimated the actual number proteins bound on the DNA before resolution.

We would like to clarify that the increase in the number of proteins bound to the DNA post-resolution highlighted by this reviewer in Fig. 1e is not consistently seen in all cases thus is not a general trend. We can confirm that there were no changes in laser power settings or scanning parameters during the course of the experiment that would artificially enhance the detection of proteins after resolution.

However, perhaps this confusion comes from the figures and the movies, where we may not have sufficiently explained that after resolution, there is a drop in force, resulting in the DNA no-longer being in the confocal imaging plane. To confirm resolution has occurred and the DNAs are still intact, the DNA is stretched and force extension curves are collected. We continue imaging while the extension curves are collected, such that we can visualise molecules bound to the resolved DNA, which do reappear once tension is applied. We have added more details in the main text and methods to explain this. Main text: *“could be observed between B1-2 or B3-4 after resolution once DNA tension was applied to move it back into the imaging plane”* and methods: *“Two metrics were used to confirm resolution: force and imaging. Once resolved, the force as measured on bead 1 and 4 dropped to essentially 0 pN. The loss of force often resulted in loss of the DNA from the imaging plane. To confirm DNA was still present, DNA force extension data was collected, such that an increase in force could confirm the presence of DNA, as well as move the DNA back into the imaging plane, for visual confirmation of DNA between the beads.”*

While we acknowledge that the exact number of proteins bound before resolution might be underestimated due to potential limitations in detection sensitivity, this does not affect the main conclusions of our study. Our key findings, particularly regarding resolution probabilities, are based on the statistics of experiments carried out under consistent conditions, with similar protein concentrations across all experiments. As we mentioned in our previous reply, the binding after braid resolution can be explained through new binding events. Importantly, the variance in the number of bound enzymes after resolution does not compromise the validity of our conclusions of resolution probabilities.

6. TOP2 α resolution of DNA braids requires ATP hydrolysis

I agree with the authors that it is of interest how ATP hydrolysis step (binding, hydrolysis, and release) is coupled to the three-gates motion and DNA transfer. As described in the manuscript: “It is however unclear, whether ATP binding is sufficient for release of the transported T segment through the C-terminal gate. We sought to test whether DNA braid resolution requires ATP hydrolysis.” , the authors intended to prove the T-segment release from C-gate would be hindered by ATPYS or AMPPNP. However, as mentioned in my previous review, “no unbraiding” does not prove the thesis that ATP hydrolysis is required for releasing of transfer DNA from C gate. The lack of decatenation could mean three possible scenarios i) failure to capture transfer DNA (as ATP analogue binding may lock the gate before DNA capture), ii) failure to pass transfer DNA through the DNA gate, and iii) failure to release the transfer DNA from the C-gate after passage through the DNA gate. One way to confirm if and where T-segment is located within the protein could be done by either establishing that the DNA molecules are still linked by after mechanically reversing the crossing (transfer-DNA captured by enzyme; case of ii and iii) or by removing protein with SDS after decatenation test to see if the braid could be separated (iii).

We appreciate the reviewer's detailed feedback regarding the ATP hydrolysis mechanism and its coupling with the three-gates motion and DNA transfer. We acknowledge the reviewer's concerns about the interpretation of the "no unbraiding" result and the potential for multiple scenarios affecting DNA release from the C-gate.

As mentioned in our previous response, our current experiments focused on testing whether ATP hydrolysis is required for braid resolution, in line with previous findings from Roca and Wang (Roca and Wang 1994, Cell 77:609-16) indicating that Top2 can decatenate plasmids without ATP hydrolysis. Our suggestion that the DNA is in the C-gate was based on this prior research. However, we fully recognize that the three scenarios proposed by the reviewer (failure to capture, failure to pass through the DNA gate, and failure to release from the C-gate) are all plausible and require further investigation.

We agree that the additional experiments proposed by the reviewer could provide clarity on the T-segment's location within the protein and its release mechanism. Given the complexity and importance of these questions, we believe that a thorough study encompassing both biochemical and biophysical approaches is necessary here. We feel that such a detailed exploration of these scenarios is beyond the scope of the current manuscript. Nonetheless, we have incorporated the reviewer's suggested three options into the discussion to highlight the ongoing debate in this area.

We have added the following stence to the manuscript: *“We did not observe DNA braid resolution when we substituted ATP for ATPyS or AMPPNP in our assays (Fig. 3a), despite the fact that TOP2α was able to bind to DNA braids (Fig. 3b). This could be due to three possible senarios: 1. Failure to capture the T-segment, 2. Failure to transfer the T-segment through the DNA-gate and 3. Failure to release the T-segment from the C-gate.”*

7. The Diffusion measurements of TOP2 α with and without ATP showed some interesting results. However, the authors should take into account the residual flow effect in the cell that the authors dismissed as the flow caused only ~3 pN drift force significantly smaller than 60 pN.

We speculated that this low force background detected on bead 4 in figure 1f could be due to residual flow, as this is one possible explanation. Given beads are moving during the collection of force extension curves, it could be caused by other factors such as turbulence caused from movement of other beads, or interference induced from moving optics. The two latter explanations would not affect the diffusion data. Furthermore, if there was residual flow during the collection of Kymos we would expect some sort of bias in localisation or direction of movement, which we did not observe. In figure 3d and e, the two examples show the initial and final locations, and direction of motion are different. In any case, we are comparing the behaviour of with and without ATP in the same system, hence our comparison is valid regardless.

8. TOP2 α resolves DNAs with multiple braids

Multiple braid resolution by different type IIA topoisomerases have been extensively investigated in previous studies at the single molecule level as listed in the references (ref 35, 36 and 38). Previous work elucidated the processivity of topo IIA on different braid geometry, chirality, and tension dependence. In this manuscript, the authors tested this under less rigorous condition where 4 braids were applied under high enzyme concentration without any crossing geometry information and a single poorly controlled force 5 ± 5 pN.

We agree with the reviewer in that the previous studies referenced (refs 35, 36, and 38) have investigated Top2 processivity on multi-braided substrates. These studies largely employed magnetic tweezers, to investigate multiple braid resolution. In contrast, our study employs optical tweezers, which offers complementary data. Optical tweezers allow for measurement at the single-molecule level, albeit with different constraints compared to magnetic tweezers. Each technique has its unique strengths and weaknesses, and it is through the integration of data from various approaches that we can achieve a more holistic view of the molecular mechanisms at play.

Therefore, we respectfully disagree with the notion that our conditions are less rigorous. Rather, we believe that our approach complements the existing body of work, providing additional insights that are only possible through the use of optical tweezers. We have now added the details regarding braid geometry at the different forces and the exact applied forces (with the deviation caused by movement and flow) in the revised manuscript to strengthen our analysis further.

9. Fig.2e: Before full decatenation (middle image), three bright dots appeared in a row between two beads (bead 4 and 3) that almost looked like bound on DNA. After decatenation (image on the right), those dots still remained and one more dot appeared on the same horizontal line next to one of the previous dots. I am sure if those images actually corresponds to the decatenation activity detected by force change below.

Yes, the reviewer is correct, resolution occurs during the confocal scan. The scan is collected one pixel at a time, one line at a time, hence the top part of the image corresponds to an earlier time point than the lower part of the image. We have included a grey rectangle in the image in 2e indicating at which point of the scan resolution has taken place.

10. Etoposide inhibits TOP2 α activity

The previous study by Le et al. (ref. 37) provided detailed mechanism of Etoposide dependent TOP2 α inhibition. At 50 μ M etoposide, reference 37 showed a minor reduction in the rate for human Top2 α compared to no etoposide as the etoposide dependent inhibition frequency was low (0.02-0.04 per turn). Interestingly, the current study showed 6/13 decatenation for single braid and 0/9 for 4 braids indicating that the inhibition frequency is >0.5 per turn. In my previous review, I asked the authors to explain what would cause such discrepancy. However, the authors response was somewhat confusing and even contradictory as below.

“As mentioned earlier, our four-braided substrate would require four TOP2a uncrossing cycles (not 2). However, we do not think that our results are inconsistent with those from Ref. 31 (Le et al. 2023). We have, therefore, stated that our results are compatible with what was reported in that study (Le et al. : page 6). Firstly, the magnetic tweezer experiments by Le et al. are carried out at 0.2 pN, not 5 pN. As shown in Figure 3, higher forces can affect TOP2a activity. Secondly, the data in Le et al. (Figure 6b, blue curve therein of their publication) does show the presence of a few small bursts of activity separated by very long pauses in the 300 s timescale, which are as long as our five-minute experimental window. Furthermore, in the Le et al. experiments, the DNA never returns to the initial length, indicating that etoposide does not allow TOP2a to fully uncross the DNA, in agreement with our results. Lastly, in the magnetic tweezer experiments by Le et al., the enzyme cannot be observed directly, and therefore, the reported activity could be the result of a multimer of TOP2a, rather than individual monomers (as suggested by Yogo et al.). “

I agree that higher forces affect Top2 α activity as shown here and in previous studies but we do not know if higher forces would also affect etoposide dependent Top2 α inhibition without further experiments. The authors seem to be speculating that force affects etoposide inhibition to a greater extent than Top2 α activity, which would be an interesting result, but would require substantially more force-dependent relaxation data to confirm.

The authors claimed that the data in Le et al. showed 300s pauses but figure 6b only showed an exemplary trace (Yeast topoll) and did not reflect an actual average pause duration.

Additionally, the enzyme concentration used in the study by Le et al. was 1 pM. Based on the study by Yogo et al, the data was highly likely taken with a single enzyme (Top2 α dimer). If multimer enzymes were present, I would expect more etoposide dependent Top2 α inhibition in terms of etoposide concentration compared to no etoposide condition. In line with this, 2 nM enzyme concentration (2000x more than one in the study by Le et al) used in the current study might explain higher inhibition frequency.

I do not find the information (“the DNA never returns to the initial length”) mentioned by the author in the study by Le et al.

We appreciate the reviewer's analysis and acknowledge the points raised regarding the discrepancies between our results and those of Le et al. (Ref. 37). We understand that our previous response may have been unclear and potentially confusing, and we would like to clarify our position.

Firstly, we agree that the higher forces applied in our experiments could influence Top2 α activity, as also noted in previous studies. However, we fully acknowledge that without direct experimental evidence, it is speculative to assert that force affects etoposide inhibition more than Top2 α activity. We intended our previous comments as potential explanations rather than definitive answers. We agree with the reviewer that further force-dependent relaxation data would be necessary to substantiate any such claims.

We also acknowledge the reviewer's point about enzyme concentration. The higher enzyme concentration used in our study could indeed explain the observed differences in inhibition frequency. We understand that, as the reviewer noted, higher concentrations might lead to more pronounced etoposide-dependent inhibition. We have amended our manuscript to include this perspective and to emphasize that our previous statements were speculative and offered as possible explanations.

Finally, regarding the observation that "the DNA never returns to the initial length," we appreciate the reviewer bringing this to our attention. Upon further review, we agree that this specific detail may not have been explicitly mentioned in the study by Le et al.

11. TOP2 α DNA braid resolution requires controlled loading to the DNA junction

This is a potentially new result, but with the additional information provided in the rebuttal and revised manuscript I can identify two reasons why the conclusion from the results obtained in this study is less convincing.

First, as mentioned in my previous comment, it would be hard to draw a conclusion from 9-12 trials with 5 min measurement window for any Poisson

process of which mean waiting time is significantly longer than 5 min. I agree with the authors that the enzyme binding at the crossings or crossing junctions should facilitate decatenation activity, particularly at high tension (5 pN) and a large crossing angle under which the diffusion of DNA crossing would be slow or limited. However, I am uncertain about how the authors drew the conclusion from the example of some small number of bound enzymes (~1nM). As shown in the study by Yogo et al. (PLOS One, 2012), the waiting time to observe a first burst of activity was ~ 100s even at 3.7 pM and no unlinking activity at 0.1 pM for 300 s cut-off time (interesting that the authors also chose the same time duration). This suggests that the reason for no decatenation for prebraided DNA in the current study could be just due to the extremely low probability to encounter a correct crossing geometry for decatenation even for Top2 α binding close to the junction.

In response to the reviewer's concerns, we want to clarify that our conclusion stating that for resolution it is important that Top2 α loads onto the braid directly comes from two different set of experiments, first the re-braiding assay in which we saw 0 resolutions out of 10 events (Fig. 3a), and second, our preloading experiments (where Top2 α is loaded before braiding) where we observed 0 resolutions out of 9 events (Fig. 3a), importantly Top2 α molecules were detected at the crosses in these experiments.

We disagree with the statement by this reviewer that the resolution events we measured would require mean waiting times longer than 5 min, our data clearly shows that Top2 α is highly efficient in resolving braids within the 5 min trials in our conditions (force and angle) when braids are generated before being exposed to Top2 α (with 9 resolutions out of 9 events tries in the presence of YOYO dye and 24 resolutions out of 28 events tried in the absence of YOYO; Fig. 2a). Therefore the absence of resolution in our preloading and re-braiding experiments demonstrates that Top2 α is unable to resolve these braids because it is unable to bind the junction in the correct configuration/geometry, which is our conclusion.

12. TOP2 α resolves right-handed crosses more efficiently than left-handed
Chiral sensing by topoisomerase IIA is likely influenced by crossing angles rather than handedness itself as described in the previous studies (Stone et al. PNAS; Charvin et al ref 35.; Neuman et al. PNAS). The study by Yogo et al (2012 PLOS One) indicated that the crossing angle at 90° diminished the chiral preferences by TOP2 α . As such, it is difficult to conclude from the current measurement without well controlled crossing geometry of DNA braids. There is no description of the crossing angles imposed in the crossing in this manuscript, which makes the results difficult to interpret and difficult to compare to previous work in which the crossing angles were estimated.

We recognize the importance of crossing geometry in influencing the activity of TOP2 α , we have now included a description of the crossing geometry used in our experiments in the revised manuscript (figure 2 d and e, and supplementary figure 3). The crossing angles used was ~65 degrees, resulting in a bend in the DNA of 115 degrees, compatible with the bend TOP2A induces in DNA (Thomson et al,

2014, Sci. Rep and Hardin et al, 2011, NAR). As the bead motion is scripted, hence the final location of the beads, the braid geometry was largely consistent. While we acknowledge that crossing angles may significantly impact TOP2 α activity and it is an interesting factor, it is important to note that our work was not designed to investigate the effects of varying crossing angles on topoisomerase resolution. Our primary objective was to observe the chiral preferences of TOP2 α under specific conditions. Testing different geometries and its effects on decatenation was not a point raised during the first round of review, and as such, we believe it constitutes a new issue that should not be introduced at this stage of the review process.

We also wish to highlight that the technical approaches used in the studies mentioned differ from our methodology. As a result, we believe that directly extrapolating results from those studies to our findings may not be appropriate. Nonetheless, at the angles tested in our experiments, we observed clear chiral preferences by TOP2 α .

13. TOP2 α DNA braid resolution is inhibited with force

As referred in the current manuscript (ref. 35 and 38), force-induced decatenation rate inhibition of topoisomerase IIA was previously investigated in detail. Based on the decatenation rate in terms of force (ref 38), the estimated waiting time for a single enzyme to decatenate would be ~32s at 5 pN and ~1000s at 10 pN indicating that the probability to decatenate at 30 pN within 5 min would be very low.

As mentioned in response to reviewer 2 comment 1 and review 1 comment 5 we have measured resolution at fixed forces of 4.8 ± 0.5 , 14.6 ± 0.5 , 24.8 ± 0.2 , 30.7 ± 0.8 and 44.8 ± 0.9 pN, and we see no difference between the resolution efficiency at 5 and 15 pN. This is new and insightful result, and highlights that one cannot extrapolate from existing data, where the rate limiting step may be different at different forces.

The reviewer assumes the force dependency can be extrapolated and the same rate limiting steps occur at forces greater than 5pN. The reviewer extrapolated using data fit in reference 38 (Seol, et al), which used human Top2a and was assumed to fit a tension dependent rate of:

$$v=v_0\exp(-\langle T \rangle \Delta x/kBT).$$

Fitted values for positive writhe (right handed) were $v_0=5.2 \text{ s}^{-1}$ and $\Delta x= 8.1 \text{ nm}$, hence at 5pN, rate would be 0.0003 cycles per second, or ~3000 seconds per cycle. Fitted values for negative writhe (left handed) were $v_0=2.2 \text{ s}^{-1}$ and $\Delta x= 3.5 \text{ nm}$, hence at 5pN, rate would be 0.036 cycles per second, or ~30 seconds per cycle. It appears the reviewer used the wrong handedness in their extrapolation, furthermore, Seol et al points out that this equation fit poorly for RH braids (positive writhe), as there are likely more than one force dependent process occurring. We have used survival analysis with our 5 minute sampling window to estimate a rate of 106–200 s per resolution and discussed this in text.

14. Cohesin inhibits TOP2 α DNA braid resolution

I agree with the assumption that TOP2 α decatenation activity would be hindered if other enzymes physically block the DNA crossing. As shown in the current study, not only Cohesin but also Condensin, which is known for enhancing TOP2 α decatenation activity in vivo (Dyson et al, EMBO 2021), could interfere with TOP2 α when they were bound to or near to crossing. However, the question is how prevalent the single braid with high crossing angle under high tension (5 pN) used in the study occur in vivo, perhaps, except for Anaphase. In particular, both SMCs could localize at the junction suggesting that the braid condition used in the current study might drive localization of SMC proteins at the crossing. Secondly, TOP2 α could decatenate 3 out of 13 single braids (12/13 Cohesin junction binding) within 5 min suggesting that TOP2 α could overcome physical barriers and decatenate on the longer time-scale of cell-cycle steps.

The high forces (5-40pN) and the crossing angles used in our study are particularly relevant to chromosome segregation during anaphase, a critical phase for ensuring accurate chromosome decatenation. Regarding the assumption that Top2 resolution is merely affected by time, as explained earlier, our data clearly show that this view is overly simplistic and that our assays using 5 min windows are fully appropriate to test resolution probabilities at different forces. These investigations are now included in the revised manuscript (see Fig.2). As we discussed earlier, our findings demonstrate that the time component alone does not explain the efficiency of decatenation under varying conditions as assumed by this reviewer.

Moreover, our results highlight that cohesin has a greater inhibitory effect on Top2 α -mediated braid resolution compared to condensin. We propose that this may reflect cohesin's higher affinity for binding to DNA braids, thereby occluding the substrate and hindering Top2 α 's ability to resolve the braids effectively. This result as well as our interpretation is a crucial, novel and interesting finding that adds to our understanding of the distinct roles these proteins play in modulating Top2 α activity during cell cycle progression.

In Summary, now that the authors have provided additional information and revised the manuscript in response the questions from the reviewers, my enthusiasm for the work has significantly declined. The majority of the results reproduce previous results, though with lower resolution, accuracy and experimental control, and the few possibly new results are open to interpretation given the experimental limitations. I cannot recommend publication in Nature Communications.

We appreciate the reviewer taking the time to provide additional feedback. However, we strongly disagree with the evaluation presented in this latest comment. The reviewer's assertion that our work primarily reproduces previous results, and with lower resolution, accuracy, and experimental control, is both unfounded and unjust. Throughout the manuscript, we have clearly demonstrated how our work advances

the field, including introducing new methodologies and providing insights that were not previously available.

The reviewer's claim that our few new results are "open to interpretation" due to "experimental limitations" is particularly troubling, as it dismisses the rigorous experimental design and controls we have implemented. Our results are well-supported by the data and have been carefully analyzed to account for any potential limitations.

The language used in this evaluation seems to go beyond constructive criticism and verges on being dismissive and unprofessional. We believe that such a tone is not conducive to the scientific discourse that peer review aims to foster. It appears that the reviewer may have approached our work with a biased perspective, focusing on discrediting rather than engaging with the substantive contributions our study makes to the field.

We respectfully request that the review process be handled with the fairness and professionalism expected in a journal of Nature Communications' stature.

Reviewer #3 (Remarks to the Author):

I am happy with how authors addressed all my comments. One thing that I spotted in new data is I would recommend double checking the trap stiffness and state explicitly how signals in 2d,e were processed. Currently authors wrote that trap stiffness was 0.3 pN/nm. The first step in 2e is ~ 2pN, which would make it ~ 6 nm and the noise less than 1nm. Is this a bootstrapped data or how do authors get such high resolution?

We acknowledge we did not provide these details in text. The data presented in figure 2 d and e (new figure 4) is down-sampled for clarity. Force data is collected at two frequencies, low frequency with time points matching bead tracking frequency (15Hz) and high frequency matching the time frequency of the traps (15000 Hz). The data is the high frequency data down sampled to 1000 Hz. This has been added to the methods: "*Force vs time curves were plotted with a down-sampling rate of 1000 Hz*" and the scripts used to create the figure detailing data handling and plotting are provided at Mendeley Data, DOI: 10.17632/vy9h9cpjyy.1.

Also, even a single braid would probably be expected to be significantly longer than that given DNA persistence length on the order of 50 nm. This might affect interpretation. How much DNA length was resolved in a single 2 pN step and is it consistent with a fraction of braids?

The measures of DNA length we have in our system are via direct visualisation and charges to the force extension curve. Direct visualisation is not sensitive enough to distinguish such small distance changes, and changes in the force extension curve between one and four braids could be confounded by increases in DNA/DNA contacts and friction as force is applied. Our best estimate for the change in length is from the FE curves of naked DNA, where the distance change from 7 to 5 pN (shown in Fig 4e) corresponds to change in length of approximately 300 bp. We can only

speculate whether this number is too high or low for incomplete resolution of 4 braids (ie 1-3 braids) under force. Previous studies on DNA topology suggest mini-circles of DNA 339 bp in length can form writhe when supercoiled (Pyne, Nature comms, 2021), suggesting DNA crosses can involve regions as short as a few hundred base pairs when under strain.

Also, what “relative force” means is unclear.

This was left behind from the previous use of relative force. This has been corrected to read just “Force (pN)”.

Answer to Reviewers' comments :

We would like to sincerely thank the reviewers for their constructive evaluation and thoughtful criticisms throughout the review process. Their feedback greatly helped us improve the manuscript, and we are pleased to see that they are satisfied with the final version.